# Evolution of the complex transcription network controlling biofilm formation in *Candida* species

**Eugenio Mancera[1]\*, Isabel Nocedal[2†], Stephen Hammel[3‡], Megha Gulati[4§], Kaitlin F Mitchell[5#], David R Andes[5], Clarissa J Nobile[4], Geraldine Butler[3], Alexander D Johnson[2,6]**

[1]Departamento de Ingeniería Genética, Centro de Investigación y de Estudios Avanzados del Instituto Politécnico Nacional, Unidad Irapuato, Irapuato, Mexico; [2]Department of Microbiology and Immunology, University of California, San Francisco, San Francisco, United States; [3]School of Biomolecular and Biomedical Science, Conway Institute, University College Dublin, Dublin, Ireland; [4]Department of Molecular and Cell Biology, University of California, Merced, Merced, United States; [5]Department of Medical Microbiology and Immunology, University of Wisconsin, Madison, United States; [6]Microbiome Initiative, Chan Zuckerberg Biohub, San Francisco, United States

**\*For correspondence:**
eugenio.mancera@cinvestav.mx

**Present address:** [†]Department of Biology, Massachusetts Institute of Technology, Cambridge, United States; [‡]The School of Computer Sciences and IT, Western Gateway Building, University College Cork, Cork, Ireland; [§]Molecular Cell, Cell Press, Cambridge, United States; [#]Center for Global Health, Centers for Disease Control and Prevention, Atlanta, United States

**Abstract** We examine how a complex transcription network composed of seven 'master' regulators and hundreds of target genes evolved over a span of approximately 70 million years. The network controls biofilm formation in several *Candida* species, a group of fungi that are present in humans both as constituents of the microbiota and as opportunistic pathogens. Using a variety of approaches, we observed two major types of changes that have occurred in the biofilm network since the four extant species we examined last shared a common ancestor. Master regulator 'substitutions' occurred over relatively long evolutionary times, resulting in different species having overlapping but different sets of master regulators of biofilm formation. Second, massive changes in the connections between the master regulators and their target genes occurred over much shorter timescales. We believe this analysis is the first detailed, empirical description of how a complex transcription network has evolved.

## Introduction

Many of the most medically relevant fungi belong to the *Candida* genus. These microbes are part of the human microbiota, but under specific circumstances — such as imbalances in components of the microbiota or suppression of the immune system of the host — they can proliferate as opportunistic pathogens and cause disease (*Calderone and Clancy, 2012*; *Turner and Butler, 2014*; *Kullberg and Arendrup, 2015*; *Romo and Kumamoto, 2020*). These diseases, which were already documented by the ancient Greeks, range from mild cutaneous disorders to systemic infections with high mortality rates (*Lynch, 1994*; *Calderone and Clancy, 2012*; *Kullberg and Arendrup, 2015*; *Nobile and Johnson, 2015*). Although they are usually studied in planktonic (suspension) cultures in the laboratory, *Candida* species, like many microbes, are often found in nature as biofilms, communities of cells associated with surfaces. For *Candida albicans*, the best studied and most clinically relevant of the *Candida* species, biofilms consist of a lower sheet of cells in the yeast form (spherical, budding cells) overlaid by a layer of filamentous cells (hyphae and pseudohyphae) and surrounded by an extracellular matrix composed of proteins and secreted polysaccharides (*Blankenship and*

*Mitchell, 2006*; *Nobile and Johnson, 2015*; *Lohse et al., 2018*). The matrix, together with specific gene expression changes within biofilms (e.g., the upregulation of drug efflux pumps), provides protection from environmental stresses including antifungal drug treatment. The ability of *C. albicans* to form biofilms has been associated both with its versatility in occupying different niches in the human host and its inherent resistance to antifungal drugs. These features are especially important for individuals with implanted medical devices, which provide substrates for biofilm formation and where often the only effective treatment is replacement of the device (*Donlan, 2001*). Biofilms also shed live yeast-form cells and thereby serve as reservoirs for further colonization in the human body (*Nobile and Johnson, 2015*).

*C. albicans* biofilm formation begins with the adhesion of yeast cells to a surface, followed by cell division and morphological differentiation to form an upper layer of filamentous cells. The biofilm matures through the secretion of the extracellular matrix (*Blankenship and Mitchell, 2006*; *Nobile and Johnson, 2015*; *Lohse et al., 2018*). In *C. albicans*, a complex transcription network regulates this process; it consists of seven 'master' transcription regulators (Bcr1, Brg1, Efg1, Flo8, Ndt80, Rob1, and Tec1) that control each other's expression and, collectively, bind to the control regions of more than a thousand target genes — around one-sixth of the total number of genes present in the genome of this species (*Figure 1*; *Nobile et al., 2012*; *Fox et al., 2015*). All of the seven regulators appear to be positive regulators that are required for normal biofilm development. Despite the complexity of the biofilm regulatory network, several lines of evidence suggest that this network originated relatively recently. For example, genes that are highly expressed during biofilm formation are enriched for genes that are relatively young, meaning that they only have a clear ortholog in species closely related to *C. albicans* (*Nobile et al., 2012*). Apart from the literature available for *C. albicans*, most of the work to understand biofilm formation in *Candida* species has been carried out with *Candida parapsilosis* (*Ding and Butler, 2007*; *Connolly et al., 2013*; *Holland et al., 2014*). *C. parapsilosis* diverged from a last common ancestor with *C. albicans* nominally 70 million years ago (*Mishra et al., 2007*; *Butler et al., 2009*). Although six of the seven master regulators of biofilm formation in *C. albicans* have clear orthologs in *C. parapsilosis*, only two of them are required for biofilm formation in the latter species (*Holland et al., 2014*). *Candida dubliniensis* and *Candida tropicalis* are more closely related to *C. albicans* (see *Figure 2*) and are also known to form biofilms (*Ramage et al., 2001*; *Silva et al., 2011*; *Pujol et al., 2015*; *Araújo et al., 2017*; *Dominguez et al., 2018*; *Kumari et al., 2018*), but the regulatory circuits that control this process are largely unknown.

To understand how the complex transcription network that controls biofilm formation evolved, we began with the seven master regulators of biofilm formation in *C. albicans* and determined whether their orthologs also controlled biofilm formation in *C. dubliniensis* and *C. tropicalis*. Using ChIP-seq, we mapped the targets of the orthologs in *C. dubliniensis*, *C. tropicalis*, and *C. parapsilosis*. A comparison of the extant networks showed that the two main components of the network, master regulators and target genes, moved in and out of the network at very different rates over evolutionary time. While the regulators moved gradually and in rough correlation with small phenotypic changes in biofilm structure, the master regulator-target gene connections changed very quickly. The large-scale changes in connections observed between closely related species did not appear to have a major impact on biofilm phenotypes, at least as monitored *in vitro*. These results suggest an evolutionary route through which complex regulatory networks could rapidly explore new network configurations (and perhaps new phenotypes) without disrupting existing functions.

## Results

### Only closely related species to *C. albicans* form complex biofilms

To understand how the transcription network that controls biofilm formation changed over evolutionary timescale, we first phenotypically characterized the biofilms formed *in vitro* by the different species of the so-called CTG clade. This clade, which includes but extends beyond *Candida* species, was traditionally named CTG due to its unusual property of decoding the CTG codon as serine instead of the usual leucine (*Figure 2*). Recently, this clade has been renamed CTG-Ser1 because other Ascomycota clades were discovered to also have unusual codon usage (*Krassowski et al., 2018*). To define an optimal growth medium for these assays, we tested biofilm formation under

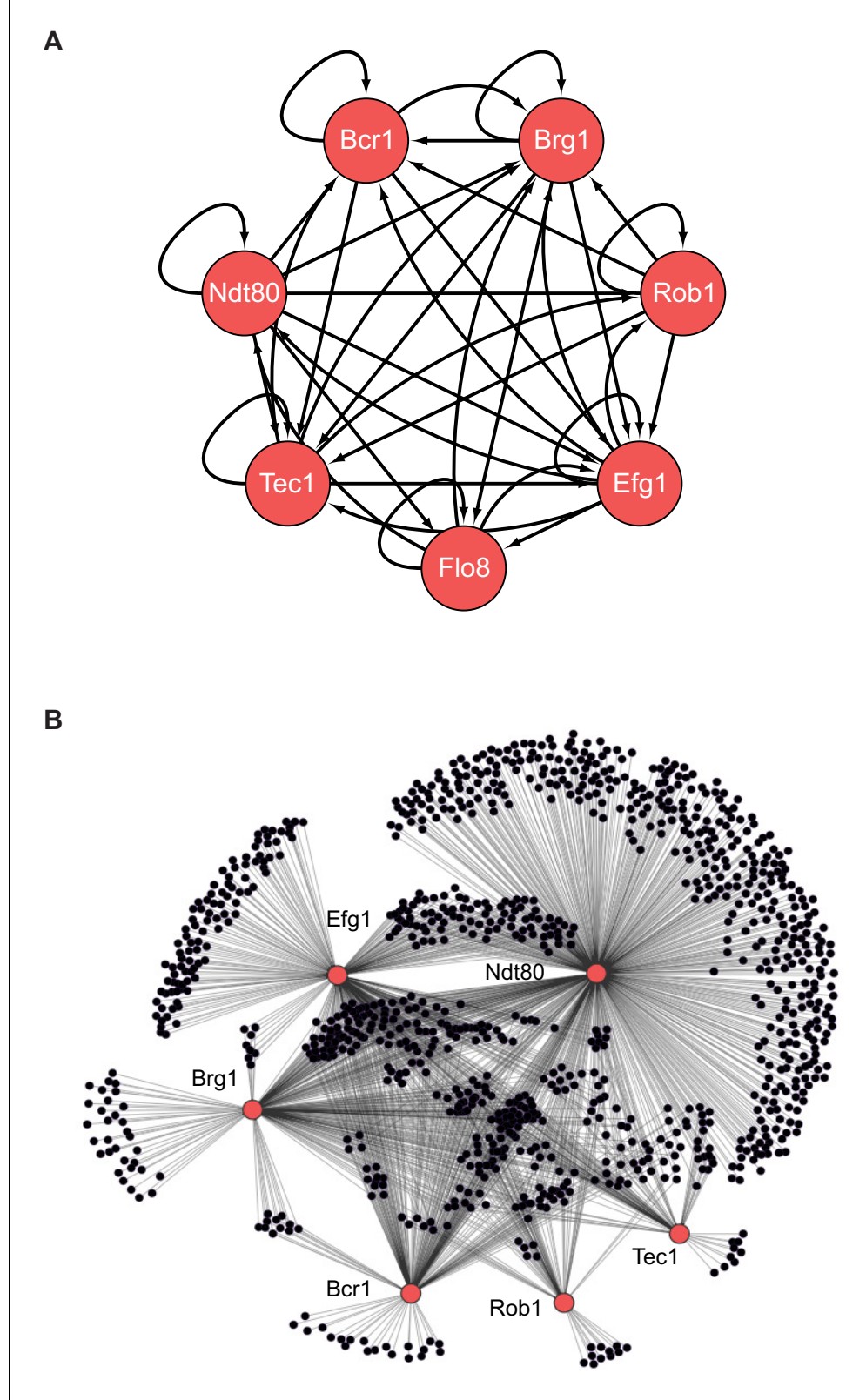

**Figure 1.** The biofilm transcription network in *Candida albicans*. (A) The seven master transcription regulators identified in genetic screens and the interactions among them as determined by ChIP-chip and ChIP-qPCR (*Nobile et al., 2012*; *Fox et al., 2015*). (B) Binding interactions (determined by ChIP-chip) between the master regulators (red) and their target genes (black). Figure adapted from *Nobile et al., 2012*. Many target genes are

*Figure 1 continued on next page*

*Figure 1 continued*

bound by more than one regulator. Note that genome-wide binding data is not available for Flo8, and thus it is missing from the larger network diagram in (B).

several conditions typically used to study biofilms of *Candida* species (*García-Sánchez et al., 2004*; *Richard et al., 2005*; *Kucharíková et al., 2011*; *Nobile et al., 2012*; *Lohse et al., 2017*). For the initial tests, we focused on *C. albicans* and the three species that are most closely related to it and that commonly inhabit humans, *C. dubliniensis, C. tropicalis,* and *C. parapsilosis* (*Figure 2*; *Turner and Butler, 2014*; *Gabaldón et al., 2016*). The estimated divergence time for *C. dubliniensis, C. tropicalis,* and *C. parapsilosis* from the last common ancestor with *C. albicans* is approximately 20, 45, and 70 million years, respectively (*Mishra et al., 2007*; *Butler et al., 2009*; *Moran et al., 2012*). *C. albicans*, *C. tropicalis,* and *C. parapsilosis* have been previously shown to form biofilms, while less is known about biofilm formation in *C. dubliniensis* (*Silva et al., 2011*; *Araújo et al., 2017*; *Dominguez et al., 2018*; *Kumari et al., 2018*). Biofilms were grown *in vitro* on silicone squares at 37°C for 48 hr with shaking and were monitored by confocal scanning laser microscopy (CSLM), as has been previously described (*Nobile et al., 2012*). We tested eight different growth media, and only Spider medium with glucose (rather than mannitol) as the carbon source allowed all four species to form thick, well-structured biofilms (*Supplementary file 1a*). Our results also showed that environmental conditions are important determinants of biofilm formation for some of these species. While *C. albicans* formed thick biofilms in all media tested, *C. tropicalis* biofilm formation, for example, depended very much on carbon source (*Supplementary file 1a*).

Given that there could be differences in the speed at which different species form biofilms, we also assessed biofilm formation as a function of time for the same four species. Biofilms were formed as described above and were monitored at seven different time points from 30 min to 96 hr after cell adhesion under the confocal microscope. Although *C. albicans* formed biofilms more rapidly, by 48 hr all four species had formed mature biofilms that did not significantly change at later time points (*Figure 2—figure supplement 1*).

Once we had defined an optimal biofilm growth medium (Spider + glucose) and time point (48 hr), we extended our analysis to other species of the CTG clade (*Maguire et al., 2013*). In addition to CSLM as described above, we monitored biofilm formation in two additional ways: we determined the biomass dry weight of biofilms formed on the bottoms of polystyrene plates, and, using a microfluidic flow cell, we continuously monitored biofilm formation by time-lapse photography using an optical microscope (*Nobile et al., 2012*; *Lohse et al., 2017*). The three methods are complementary: biomass determination is a quantitative method that reduces biofilm formation to a single number, confocal microscopy is qualitative, but allows detailed characterization of the structure of the biofilm, and the microfluidic assay reveals biofilm formation in real time under a defined flow; the flow rate was adjusted to mimic that of an average catheter implanted in a vein (*Gulati et al., 2017*; *Lohse et al., 2017*). Because some of the species we tested are known to grow poorly at 37°C, we performed the assays at 30°C for those species (*Kurtzman et al., 2011*). Although not all species were tested in the three assays, overall, of the 15 species analyzed, those closest to *C. albicans* formed the thickest biofilms and, in general, the greater the phylogenetic distance from *C. albicans* the thinner the biofilm formed (*Figure 2*, *Figure 2—figure supplement 2*). Only the two species closest to *C. albicans* (*C. dubliniensis* and *C. tropicalis*) formed biofilms that are structurally very similar to *C. albicans* biofilms, with a basal layer of yeast cells underlying a thick layer of filamentous cells (hyphae and pseudohyphae). Biofilms formed by *C. parapsilosis* appeared similar at low resolution, but a closer examination showed that the layer of filamentous cells is composed largely of pseudohyphae rather than a mixture of true hyphae and pseudohyphae (*Figure 2B*). Under these conditions, *Lodderomyces elongisporus* formed thinner biofilms composed only of yeast cells. Moving further away from *C. albicans*, *Spathaspora passalidarum*, *Meyerozyma guilliermondii,* and *Clavispora lusitaniae* form even thinner biofilms, while *Scheffersomyces stipites*, *Debaryomyces hansenii*, *Metschnikowia bicuspidate*, *Hyphopichia burtonii,* and *Candida tenuis* did not form biofilms under the conditions we tested; only a few cells were observed adhering to the surface (*Figure 2*, *Figure 2—figure supplement 2*). We also performed CSLM assays in an additional medium (RPMI) with a selection of the species, and the results generally agreed with those described above (*Figure 2—figure supplement 2*). Our results with the microfluidic assays showed a similar trend: only those species

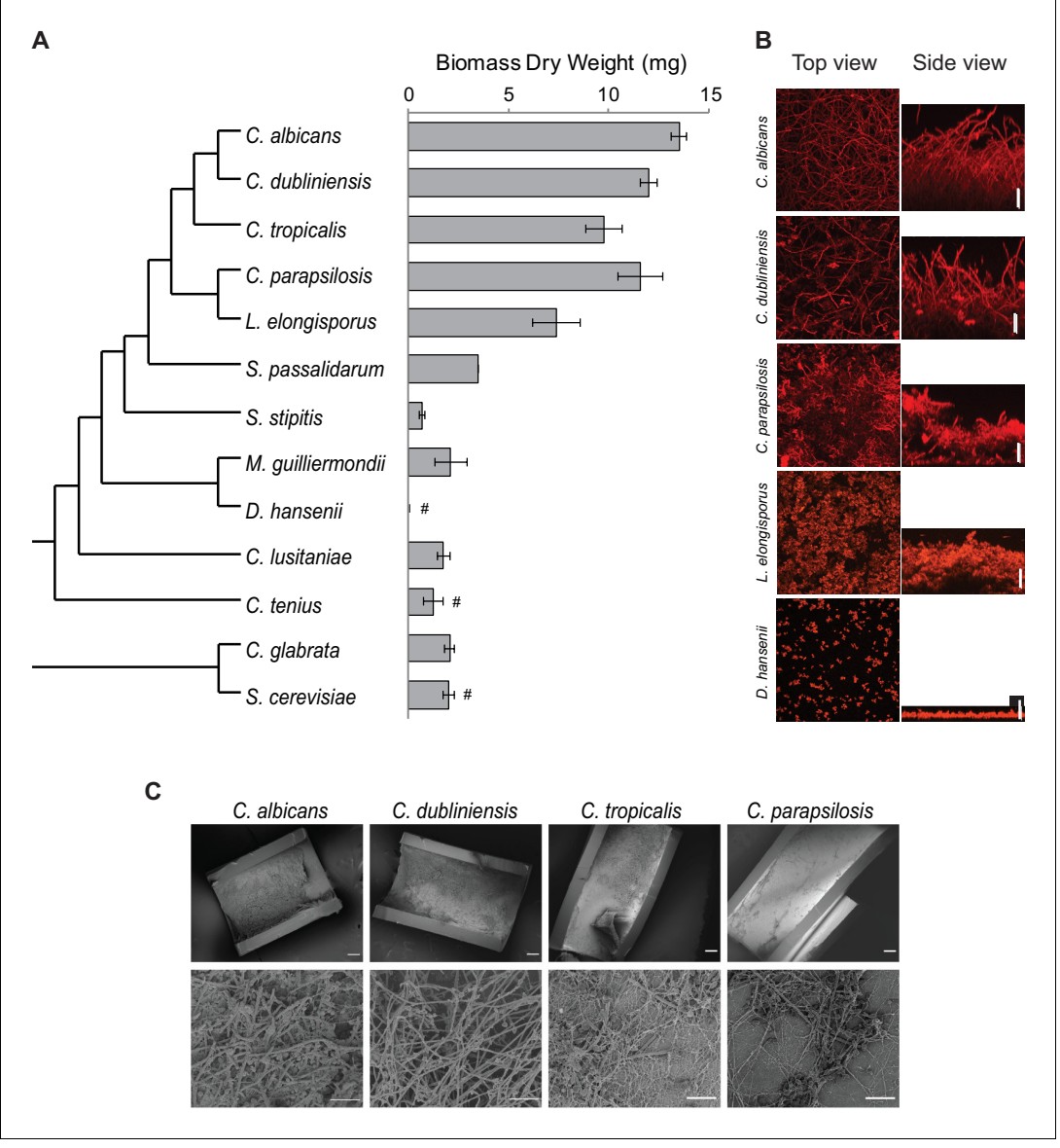

**Figure 2.** Diversity in biofilm formation across fungal species. (**A**) Biofilm biomass dry weight was determined for different fungal species grown on the bottoms of polystyrene 6-well plates in Spider 1% glucose medium at 37°C for 48 hr. The mean and standard deviation were calculated from five replicates. Hashtags denote species that do not grow well at 37°C and for which biofilms were grown at 30°C. The cladogram to the left shows the phylogenetic relationship of the species (*Byrne and Wolfe, 2005*; *Maguire et al., 2013*). All species analyzed belong to the CTG-Ser1 clade apart from *C. glabrata* and *S. cerevisiae*. (**B**) Morphology of biofilms formed by five representative CTG clade species visualized by confocal scanning laser microscopy. *C. tropicalis* biofilm morphology is similar to that of *C. albicans* and *C. dubliniensis* as shown in *Figure 2—figure supplement 2* and *Figure 3—figure supplement 1*. Biofilms were grown as described above, but on the surfaces of silicone squares. Scale bars represent 50 μm. (**C**) Biofilm formation by *Candida* species in an *in vivo* rat catheter model (*Andes et al., 2004*). Biofilms were grown for 24 hr and were visualized by scanning electron microscopy. Two magnifications are shown in the lower and upper panels for each species, and the scale bars represent 20 and 100 μm, respectively. Micrographs of *C. albicans* were adapted from *Dalal et al., 2016*, but were obtained as part of the same set of experiments performed in parallel.

The online version of this article includes the following figure supplement(s) for figure 2:

**Figure supplement 1.** Time course of biofilm formation of *C. albicans* and its three most closely related *Candida* species.

**Figure supplement 2.** Biofilm formation by CTG species.

*Figure 2 continued on next page*

*Figure 2 continued*

**Figure supplement 3.** Biofilm formation in a microfluidic device by different CTG clade species was assayed as previously described (*Gulati et al., 2017*).

---

that are phylogenetically closest to *C. albicans* were able to rapidly form biofilms under flow conditions in the microfluidic device (*Figure 2—figure supplement 3*).

As a reference, we also characterized biofilm formation in two other ascomycetous yeast species that lie outside the CTG clade, *Candida glabrata* and *Saccharomyces cerevisiae*. Although not closely related to the CTG clade (despite its name), *C. glabrata* is an important opportunistic human pathogen, while *S. cerevisiae* is used extensively in the food and beverage industries and is a widely employed model organism. As can be seen in *Figure 2*, neither of these species formed biofilms that resembled those formed by *C. albicans* and its close relatives in the assays and conditions that we tested.

To assess whether the results observed *in vitro* can be recapitulated *in vivo*, we tested the ability of *C. albicans, C. dubliniensis, C. tropicalis,* and *C. parapsilosis* to form biofilms in the rat catheter model, a well-established *in vivo* biofilm model (*Andes et al., 2004*). All four species were able to form biofilms, although the biofilms formed by *C. albicans* and *C. dubliniensis* were considerably thicker and more filamentous (*Figure 2C*). These results agree with previous *in vivo* characterizations performed for *C. albicans* and *C. parapsilosis* (*Nobile et al., 2012*; *Connolly et al., 2013*) and provide new information on the intermediate species.

In summary, our results show that the ability to form biofilms that resemble those of *C. albicans* is limited to its most closely related species. In terms of biomass, there is a sharp drop off outside *C. parapsilosis* while, in terms of biofilm structure, only *C. dubliniensis* and *C. tropicalis* form biofilms similar to those of *C. albicans,* in terms of all three morphological cell types being represented. Of all the species studied, *C. albicans* biofilm formation is the most rapid and most robust to environmental changes as it formed similar biofilms in all media tested (*Supplementary file 1a*, *Figure 2—figure supplement 2*); moreover, the biofilms formed by this species are the most stable to physical manipulation (results not shown). When interpreting these results, it is important to note that the biofilm assays employed here have been developed for *C. albicans* and its closely related species. Therefore, it is possible that our observations may not only reflect differences in the intrinsic ability to form biofilms, but could also be due to changes in the ways the species have adapted to the different environments they inhabit. The species that were not observed to form biofilms in our study could, in principle, be able to form biofilms in other conditions, but, to our knowledge, there is no evidence for this even in well-studied species such as *S. cerevisiae*.

## The regulatory core of the biofilm transcription network changed gradually over time

To gain insight into the evolutionary changes that occurred in the transcription network that controls biofilm formation at a molecular level, we first studied the function of the seven master regulators of the *C. albicans* network (*Figure 1A*). Given the phenotypic results described above, we centered the analysis on *C. albicans, C. dubliniensis, C. tropicalis,* and *C. parapsilosis*. All four of these species are common in humans (*Turner and Butler, 2014*), and the first three form similar structural types of biofilms. As described above, *C. parapsilosis* also forms biofilms, but its biofilms show more pronounced differences. All seven master regulators of the network in *C. albicans* (*Figure 1*) have clear orthologs in the other three closely related species, with the exception of Rob1. Rob1 has a patchy phylogenetic distribution, with syntenic orthologs present in *C. albicans*, *C. dubiniensis,* and *C. tropicalis*, but apparently absent from *C. parapsilosis* and closely related species. However, Rob1 orthologs are present in other more distantly related CTG species, which supports the hypothesis that Rob1 was either lost or was evolving sufficiently rapidly in the *C. parapsilosis* lineage that it cannot be recognized (*Maguire et al., 2013*).

To test whether the orthologs of the *C. albicans* master regulators are involved in biofilm formation in the other species, we generated gene deletion knockouts in *C. dubliniensis* and *C. tropicalis*. The knockouts in *C. albicans* and *C. parapsilosis* had been previously generated as part of large transcription regulator deletion projects (*Homann et al., 2009*; *Holland et al., 2014*). To make the

knockouts in *C. dubliniensis* and *C. tropicalis,* we used amino acid auxotrophic strains and employed a gene knockout strategy similar to that previously used for *C. albicans* and *C. parapsilosis* (*Mancera et al., 2019*).

The ability of the different gene knockout strains to form biofilms was monitored by biomass dry weight determination and CSLM. As can be observed in *Figure 3* and *Figure 3—figure supplement 1*, all seven master regulators identified in *C. albicans* were also required for biofilm formation

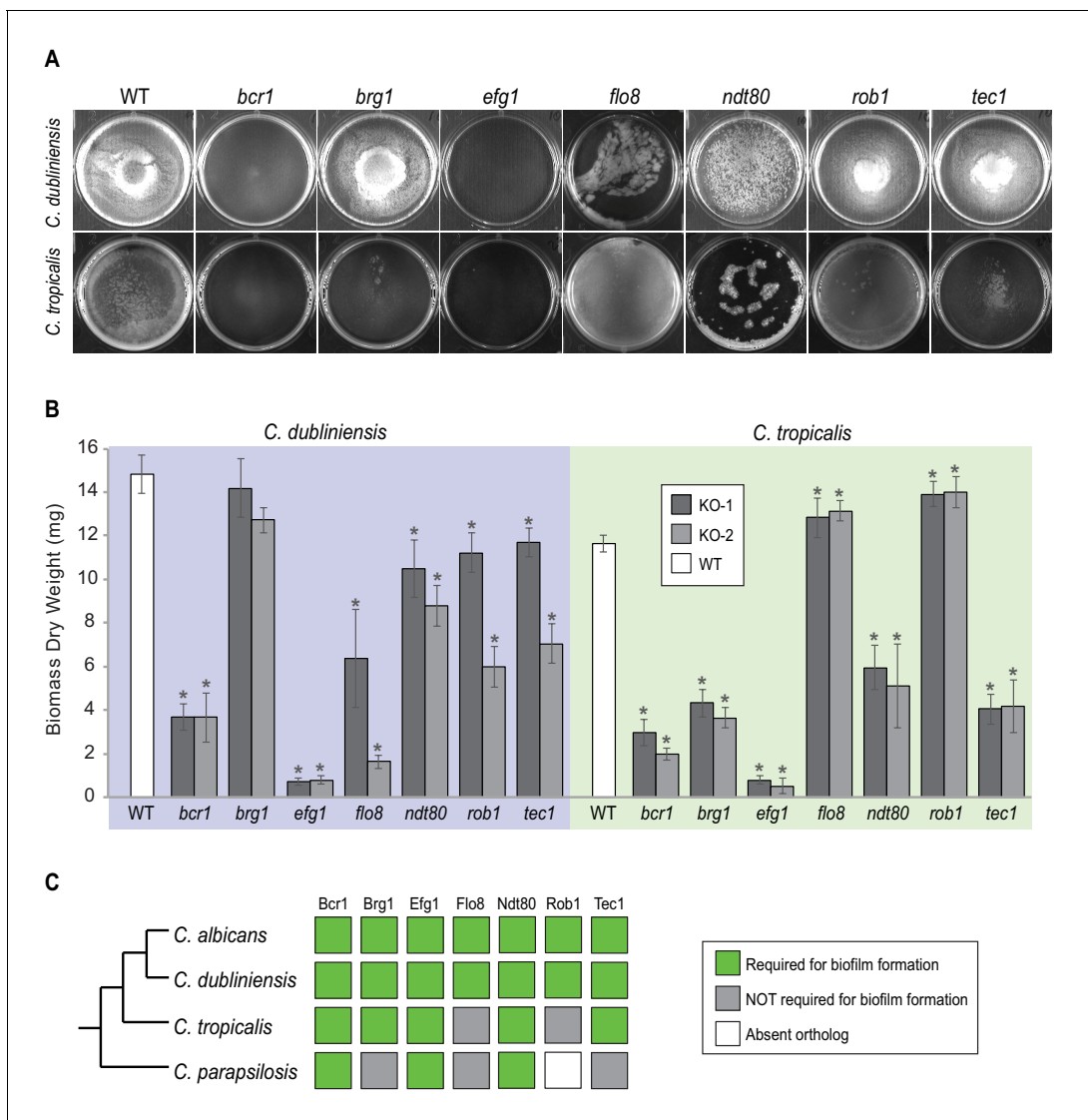

Figure 3. Roles of orthologs of the seven *C. albicans* master regulators in biofilm formation. (A) Phenotypic characterization of biofilms formed by the gene deletion knockouts of orthologs of the seven master *C. albicans* biofilm regulators. Images show biofilms grown on the bottoms of polystyrene 6-well plates in Spider 1% glucose medium at 37°C for 48 hr. (B) Dry weights of biofilms formed by the gene deletion mutants grown as described in (A). The means and standard deviations were calculated from five replicates for two independent gene deletion knockout isolates (KO-1 and KO-2). Asterisks denote statistically significant different weights when compared to the corresponding parental strain using a Student's two-tailed paired *t* test (p<0.05). Although the dry weight of the *C. dubliniensis brg1* mutant is not statistically different from that of the wildtype, detailed analysis of this mutant by confocal scanning laser microscopy showed a clear biofilm formation defect (*Figure 3—figure supplement 1*). (C) Summary diagram showing the conservation of the seven master regulators in biofilm formation across the three most closely related species to *C. albicans*. The data for *C. albicans* was obtained from *Nobile et al., 2012*; *Fox et al., 2015*, and that for *C. parapsilosis* from *Holland et al., 2014*.

The online version of this article includes the following figure supplement(s) for figure 3:

Figure supplement 1. Morphology of biofilms formed by the gene deletion mutants of the biofilm regulators in *C. dubliniensis* and *C. tropicalis* visualized by confocal scanning laser microscopy .

in *C. dubliniensis*. The results were different for *C. tropicalis*; here, only five of the seven were required, with Rob1 and Flo8 appearing dispensable for biofilm formation under our laboratory conditions. The biofilms formed by the *rob1* and *flo8* deletion mutants in *C. tropicalis* were actually slightly heavier and the hyphal layer was denser than biofilms formed by the parental (wildtype) strain, suggesting that these two regulators may have assumed a subtle inhibitory role in this species. For *C. parapsilosis,* it was previously shown that, of the seven master biofilm regulators in *C. albicans*, only Bcr1 and Efg1 are indispensable for biofilm formation. The Ntd80 deletion in *C. parapsilosis* fails to form biofilms but also has a general growth defect, making it difficult to ascertain its precise role. Previous work also established that deletion of *CZF1*, *UME6*, *CPH2*, *GZF3*, and *ACE2* all exhibit biofilm-specific defects in *C. parapsilosis* but not in *C. albicans* (*Holland et al., 2014*). Overall, our results, together with previous observations, show that the group of master regulators underlying biofilm formation in *C. albicans* is different than in other *Candida* species with the degree of difference roughly paralleling their evolutionary distance from *C. albicans*.

## Target genes of the master biofilm regulators differ greatly among *Candida* species

To determine how the binding connections between the biofilm master regulators and their target genes have changed over evolutionary time, we determined genome-wide protein-DNA interactions of the master regulators in *C. albicans*, *C. dubliniensis*, *C. tropicalis*, and *C. parapsilosis* by chromatin immunoprecipitation followed by next-generation sequencing (ChIP-seq) (*Johnson et al., 2007*). To this end, we tagged each of the regulators with a Myc epitope tag that can be immunoprecipitated using a commercially available antibody. This strategy has the advantage that a single antibody with the same affinity could be used in all experiments, and control experiments could be performed with untagged strains. All the ChIP-seq experiments were performed in mature biofilms grown for 48 hr at 37°C in the optimal medium described above. For unknown technical reasons, not all regulators could be immunoprecipitated in all species. Of all the ChIP-seq experiments performed, protein-DNA interactions could be reliably determined for 18 regulators across the four species (*Supplementary file 1b*). We believe that this coverage, although not complete, is more than sufficient to uncover the general trends of biofilm network evolution. In the following sections, we consider how the biofilm network (defined as the set of master regulators and connections between them and their target genes) differs among *Candida* species and what these differences reveal about the evolution of the network.

To compare gene targets between species, we assigned each ChIP occupancy site to the ORF with the nearest downstream start codon. To compare the changes in master regulator-target gene connections across species, we first examined the target genes that are conserved across the species. The overall percentage of one-to-one orthologs between the four species ranged from a high of 91% between *C. albicans* and *C. dubliniensis* to a low of 74% between *C. tropicalis* and *C. parapsilosis* (*Maguire et al., 2013*). If we consider only the one-to-one orthologs, it becomes clear that the connections between master regulators and conserved target genes vary greatly across these species (*Figure 4*). Between the two most closely related species (*C. albicans* and *C. dubliniensis*), the master regulator that showed the highest conservation of target gene connections was Ndt80, but the overlap was only about 50% (634 of 1,297). Between *C. albicans* and *C. parapsilosis*, this value drops to about 26% (722 of 2,725). Although only 12% (371 of 3,016) of Ndt80 target gene connections are common to all four species, the overlap for each species pair is larger than expected by chance (hypergeometric test, p<0.05). This conclusion does not necessarily mean that selective constraints preserve such overlaps over evolutionary time. Although selection could be responsible, drift is also a possible explanation, particularly given the phylogenetic proximity of the species. The other master regulators show an even lower degree of conserved regulator-target gene connections. For example, Rob1 shares only about 12% (2 of 17) of its target gene connections between the two most related species, *C. albicans* and *C. dubliniensis,* but the overlap is still greater than expected by chance (hypergeometric test, p<0.05).

Our results also show that each species has target genes in its biofilm network that lack orthologs in the other three species. In *C. albicans*, 26% of the genes bound by at least one of the biofilm master regulators do not have orthologs in the other three species compared with 19% for the genome as a whole. This analysis extends the previous observation that genes that are upregulated during biofilm formation in *C. albicans* are often 'young' genes, that is, genes that lack orthologs in related

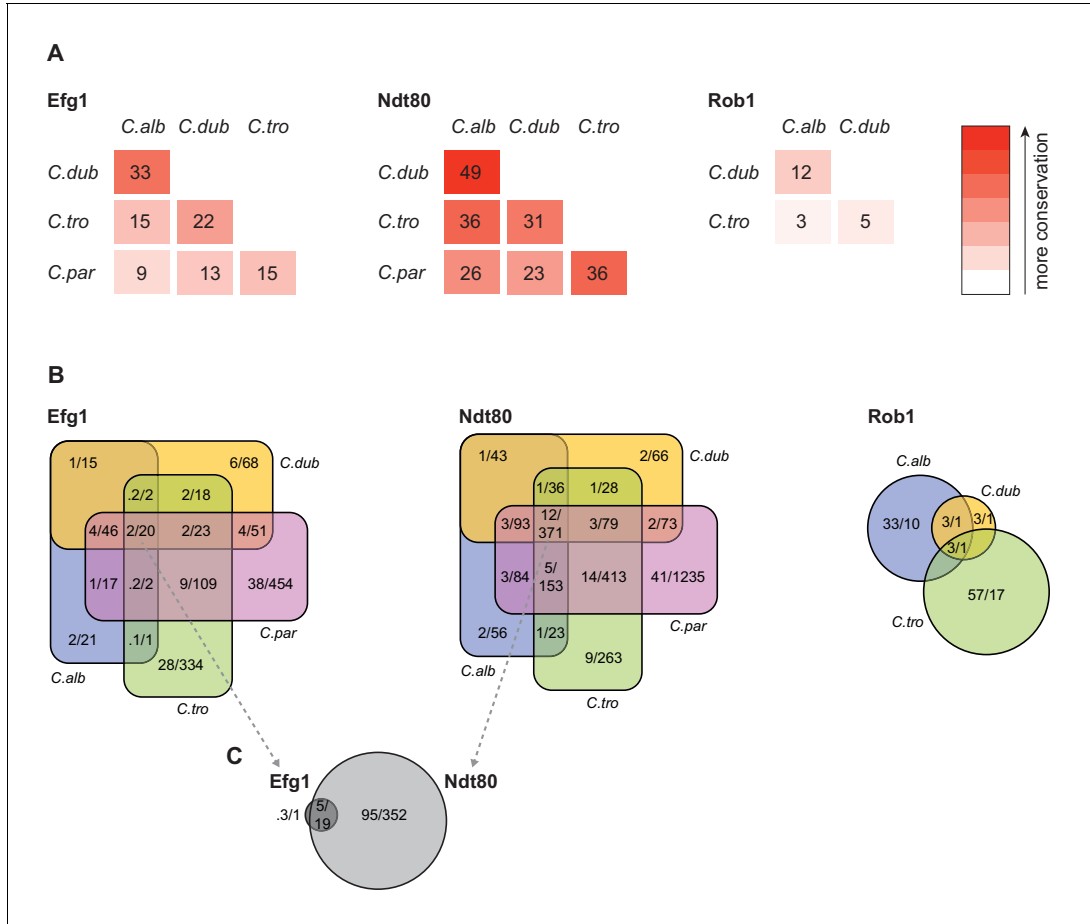

**Figure 4.** Connections between master regulators and target genes are highly divergent across species. (**A**) Pairwise comparison of shared target genes for Ndt80, Efg1, and Rob1 between species. Target genes were determined by ChIP-seq as detailed in Materials and methods. The numbers represent the percentage of overall target genes conserved between each pair of species considering only genes that have orthologs in the two species. Note that Rob1 is absent in *C. parapsilosis* (***Maguire et al., 2013***). (**B**) Venn diagrams depicting the overlap of regulator-target gene connections across species, considering only genes that have orthologs in all four species for Efg1 and Ndt80 and considering genes that have orthologs in *C. albicans*, *C. dubliniensis*, and *C. tropicalis* for Rob1. Numbers in each section of the diagrams represent the percentage of master regulator-target gene connections, with the total number of connections for each regulator set at 100%, and the gross number of target genes. Note that for Efg1 and Ndt80 the size of the color sections does not correspond to the percentage. (**C**) Venn diagram depicting the overlap between target genes of Efg1 and Ndt80, considering only target gene orthologs that are present in all four species. As in (**B**), numbers represent the percentage of master regulator-target gene connections and gross number of target genes. The diagram indicates that, for genes that are targets of Efg1 and Ndt80 in all species, most Efg1-target gene connections are also Ndt80-target gene connections, even though the target genes themselves are different across species.

The online version of this article includes the following figure supplement(s) for figure 4:

**Figure supplement 1.** Efg1 and Ndt80 target gene conservation between species when using different criteria to identify targets.

**Figure supplement 2.** Ndt80 binding throughout the genome in biofilms grown using different media.

species (***Nobile et al., 2012***). We observed a similar enrichment of unique genes in the biofilm network of *C. parapsilosis*, but not for *C. dubliniensis* and *C. tropicalis*; in the latter two species, the fraction of non-orthologous genes in the biofilm network was approximately the same as that observed for the whole genome. These observations indicate that the biofilm networks of *C. albicans* and *C. parapsilosis* have been more dynamic in recent evolutionary time than those of the other two species.

## Differences in master regulator-target gene connections are robust to different methods of comparison

We considered the possibility that our results could be skewed by false-positive signals intrinsic to ChIP-seq experiments, even when proper controls are used (*Chen et al., 2012*). To deal with this potential problem, we employed two additional criteria for increasing stringency in our analysis of transcription networks (*Nocedal et al., 2017*). First, we filtered the regions identified as enriched in ChIP signal to include only those regions that also had a high-scoring regulator binding motif in the intergenic region. As discussed below, the binding motif for some of the master regulators does not vary significantly across the species we studied. Second, we also incorporated gene expression data to further filter the gene targets to those that (1) have ChIP enrichment, (2) show the presence of a regulator binding motif in the intergenic region, and (3) whose expression changes under biofilm-forming conditions. Although the gross number of regulator-target gene connections decreased as the stringency of the filtering criteria increased, the high proportion of differences across *Candida* species described above did not significantly change (*Figure 4—figure supplement 1*).

Another potential caveat that could confound our analysis concerns our conclusions being based on the specific conditions under which we induced biofilm formation. To test whether this concern is significant, we performed the ChIP-seq experiments for Ndt80 in *C. albicans*, *C. dubliniensis,* and *C. tropicalis* in biofilms grown in an alternative growth media. In all species, the binding intensities of the regulator in all the intergenic regions of the genome were highly correlated between the two media tested (*Figure 4—figure supplement 2*). This finding indicates that there are no significant changes in Ndt80 binding when the growth media is modified.

## High connectivity of the biofilm network is observed in all four species

The initial characterization of the biofilm transcription circuit in *C. albicans* showed that many target genes were directly connected (by binding) to more than one master regulator (*Nobile et al., 2012*), and this general feature of the network architecture is observed across the *Candida* species studied here, despite the low conservation of individual regulator-target gene connections (*Figure 4A*). Perhaps the most notable example is seen by comparing the target genes of Ndt80 and Efg1 across the four species. The set of target genes of Ndt80 is considerably larger, but over 75% of the Efg1 target genes are also Ndt80 targets in all four species. In addition, the binding motifs of these two regulators are enriched in each other's binding locations in all four species studied (*Figure 5A*). These observations indicate that, in all species, Efg1 binds in conjunction with Ndt80 even though the target genes of the regulator combination differ greatly across species. The association between Efg1 and Ndt80 agrees with previous planktonic ChIP-seq experiments of Efg1 performed in *C. parapsilosis* where the most enriched binding motif found was that of Ndt80 (*Connolly et al., 2013*). The DNA-binding motifs of these two regulators have also been shown to co-occur in the binding regions of Sfl1 and Sfl2, two regulators of filamentation in *C. albicans* (*Znaidi et al., 2013*). Taken together, these results suggest that the Efg1-Ndt80 association is ancient with respect to the *Candida* species studied here and that it remains preserved across them despite large species-to-species differences in the target genes bound by the two regulators. Analysis of the other master regulators indicates that combinational control of target genes is very common in all species, although the other examples do not seem as deeply conserved as the Efg1-Ndt80 example.

In terms of overall network structure across species, we also examined whether, as is the case for *C. albicans*, the master regulators bind to their own control region as well as those of the other master regulators. The extent of conservation of the binding connections between one master regulator and the others varies from regulator to regulator (*Supplementary file 1c*), with Ndt80 showing the highest conservation. In all four species, Ndt80 binds to its own control region as well as those of all six other regulators, with the exception of the control region of *FLO8* in *C. albicans* and *ROB1* in *C. dubliniensis*. Efg1, Brg1, and Tec1, in that order, follow Ndt80 in their degree of connection conservation, with Rob1 and Bcr1 having the least conserved set of connections. However, we note that the binding data for these two regulators is also the least complete (*Supplementary file 1b*). Ndt80 and Efg1 each bind to their own control regions in all the four species analyzed, while Brg1 exhibits this interaction only in *C. albicans* and *C. dubliniensis*. The binding of the other regulators to their own upstream intergenic regions appears less conserved. Overall, our findings show that high connectivity between the master regulators is conserved in the four *Candida* species analyzed. Thus,

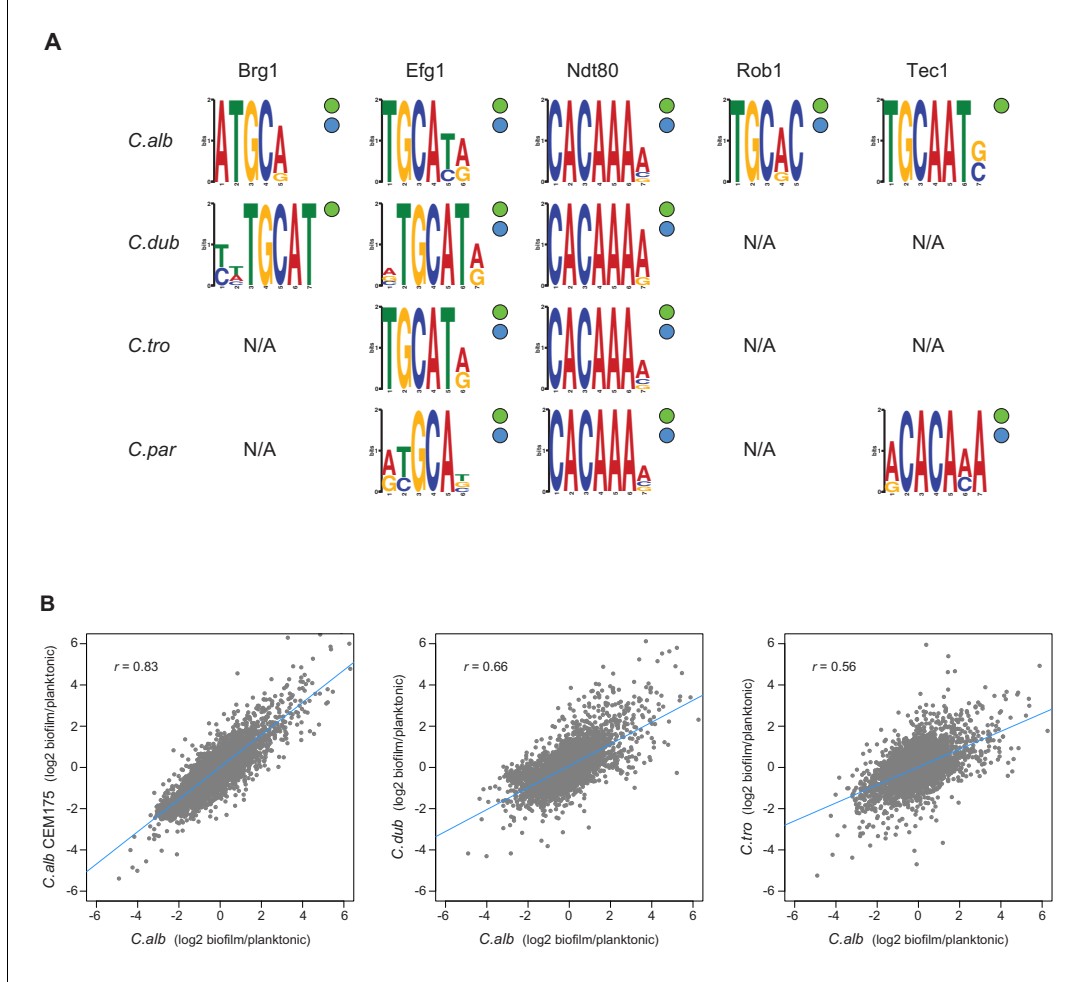

**Figure 5.** Master regulators retain their DNA-binding specificity while there is considerable variation in gene expression across species. (**A**) Logos of the most enriched motif in the binding locations of the different master regulators, determined by ChIP-seq across species. The two circles to the right of each logo show whether the Efg1 (green circle) or Ndt80 (blue circle) previously known motifs are enriched in each set of regulator binding locations. (**B**) Pairwise comparison of transcription profiles under biofilm-forming conditions (a time point of 48 hr) of *C. albicans* against *C. dubliniensis* and *C. tropicalis*. As a reference, the comparison between two isolates of *C. albicans* is shown in the left panel. Biofilm-specific expression changes were calculated comparing gene expression between biofilm and planktonic growth conditions in the same media. Linear regressions are shown in blue for each comparison.

despite the extensive changes in the network across species, the high connectivity among regulators remains a structural feature of the network in each species.

## Changes in master regulator-target gene interactions are due to changes in *cis*-regulatory regions

A possible mechanistic explanation for the high rates of change among the target genes of the biofilm network is that these changes are due to modifications in the *trans* components of the network, for example, changes in the DNA-binding specificity of a master regulator. This type of change has the potential to dramatically change the network over relatively short evolutionary timescales. To explore this possibility, we examined the ChIP-seq binding data for motifs recognized by the master regulators. Performing *de novo* motif searches, we found that the enriched DNA-binding motifs found in the binding regions of the biofilm master regulators were very similar across the four species we examined (*Figure 5*). Moreover, the *de novo* generated DNA-binding motifs for Efg1 and Ndt80 are similar to those previously reported for their orthologs in *C. albicans* and other species (***Nobile et al., 2012***; ***Nocedal et al., 2017***).

As an additional test of whether the DNA-binding specificity of the regulators changed over the evolutionary time considered here, we used the *de novo* motifs generated from the ChIP-seq data for Efg1 or Ndt80 in each species and identified the occurrence of these motifs in the upstream regions of the orthologous genes from the other species. The overlap of potential targets between species varied from 51 to 100% depending on the exact motif used, but was never low enough to account for the differences determined directly from the ChIP-seq experiments (*Figure 4*). For example, the lowest overlap observed when using motif scoring was of 51% for the Efg1 motifs in *C. dubliniensis* compared with *C. parapsilosis,* while the actual gene target overlap of ChIP-seq data for this regulator was only 13% (*Figure 4*). In other words, the large differences in regulator-target interactions across species observed in the ChIP-seq data cannot be accounted for by slight differences in the binding motifs generated in each individual species. Although we cannot fully disregard that small differences in binding affinity contribute to differences in master regulator-target gene connections observed across the species, overall, all the analyses show that the DNA-binding specificity of at least some of the master regulators has not changed significantly across these species. This conclusion is further supported by analysis of a *C. albicans–C. dubliniensis* hybrid, as described below.

## Biofilm-specific gene expression changed rapidly over evolutionary time

A strong prediction of the vast number of species-to-species differences in master regulator-target gene connections documented above is that the genes transcriptionally induced during biofilm formation should differ substantially among species. To test this prediction, we generated genome-wide transcription profiles of *C. albicans*, *C. dubliniensis,* and *C. tropicalis* under biofilm-forming conditions. To reveal biofilm-specific changes, we compared these profiles to expression data obtained when the species were grown in suspension cultures in the same medium. As a reference, we also performed the biofilm expression profile in a second *C. albicans* isolate. As seen in *Figure 5B*, the pairwise differences in the transcription profiles across the three species are significant and reflect their phylogenetic position: the further apart the two species are from one another, the less correlated their transcription profiles are. If we use a lax cutoff of twofold over/underexpression to define genes that change their expression during biofilm formation, the overlap between pairs of species is relatively low. For example, only 29% of the genes that change their expression during biofilm formation are shared between *C. albicans* and *C. dubliniensis*, and only 24% are shared between *C. albicans* and *C. tropicalis*. The overlap of differentially expressed genes between the two *C. albicans* isolates was 48%. This result is consistent with previous work showing that clinical isolates differ in their biofilm-forming abilities (*Hirakawa et al., 2015*; *Huang et al., 2019*). As noted in the 'Discussion', we believe that many of the differences among clinical strains arose after the *C. albicans–C. dubliniensis* branch.

To test whether the large differences in the gene expression profiles were specific to the media conditions used, we also performed the transcription profiles in different media, namely Spider for *C. albicans* and *C. dubliniensis*, and RPMI for *C. albicans* and *C. tropicalis*. In the alternative media, the conservation between the sets of genes that changed their expression at least twofold during biofilm formation is even lower, 22 and 11%, respectively, for *C. albicans* and *C. dubliniensis*, and *C. albicans* and *C. tropicalis*. Despite these major differences, all the interspecific pairwise overlaps are greater than would be expected by chance (hypergeometric test, p<<0.05), although we cannot distinguish if this is due primarily to selection or to the shared ancestry of their promoter sequences. Overall, the low degree of conservation in genome-wide gene expression agrees well with the low conservation of regulator-target gene connections across species described above.

To further understand the relationship between gene expression and the binding of the biofilm regulators in each species, we assessed whether differentially expressed genes were directly bound by the biofilm regulators. Considering genes that change their expression at least twofold during biofilm formation in the same conditions in which the ChIP-seq experiments were performed, the fraction of these genes bound by one or more regulators ranges from 30% in *C. albicans* to 51% in *C. tropicalis*. Combining our data with previous transcriptional profiling experiments performed in *C. parapsilosis* during biofilm formation (*Holland et al., 2014*), we estimated that 67% of differently expressed genes in this species were bound by at least one of the biofilm regulators under the experimental conditions tested. The overlap between differential gene expression and regulator binding in all species is larger than what would be expected by chance (hypergeometric test,

p<<0.05), suggesting that direct binding is an important factor in gene regulation in the biofilm network.

DNA binding of a regulator is not expected to always produce a change in mRNA production, but we did observe a correlation between these two properties. Pairwise comparisons of *C. albicans*, *C. dubliniensis,* and *C. tropicalis* showed that most genes that are expressed differentially between species (as defined above) are genes whose intergenic region is bound by at least one regulator in one species, but not in the other species. This association is statistically significant (Fisher's two-tailed exact test, p<<0.05) again suggesting that the majority of differences in biofilm-specific gene expression between species can be explained by differences in the *cis*-regulatory sequences of target genes that alter binding of the regulators.

## Analysis of an interspecies hybrid independently supports the conclusions from the species-to-species comparisons

Many challenges exist in mapping and comparing regulator-target gene connections in transcription networks between yeast species and, more generally, between any species (*Chen et al., 2012*). These difficulties include technical issues such as differential nucleic acid recovery and signal-to-noise ratios, which can vary considerably from one species to the next. However, probably the most difficult problem to circumvent arises from different species having different physiological responses to the same external environment. For example, 30°C could be the optimal temperature for one species but might induce a stress response in a closely related species. Therefore, a network comparison between these species at 30°C might be dominated not by evolutionary changes in the transcription circuitry per se but simply by the fact that only one species has induced a stress response. This problem can be overcome to a large extent by creating and analyzing interspecies hybrids, where the genomes of two different species are present in the same cell and thus exposed to the same physiological state (*Wilson et al., 2008*). This approach, which can only be carried out between closely related species, specifically reveals the *cis*-regulatory changes that have accumulated between the two genomes since the species last shared a common ancestor.

We took advantage of the fact that it is possible to mate *C. albicans* and *C. dubliniensis* (each diploid) to generate tetraploid hybrids (*Pujol et al., 2004*). These hybrids form biofilms similar to those formed by *C. albicans* (*Figure 6—figure supplement 1*). We performed ChIP-seq of Ndt80 in this hybrid, immunoprecipitating the *C. albicans* Ndt80 protein in one set of experiments and the *C. dubliniensis* Ndt80 in another set. The results showed that — in the hybrid — the target genes bound by the *C. albicans* Ndt80 and the *C. dubliniensis* Ndt80 were highly correlated, similar to two biological replicates carried out in the same species (*Figure 6A*). In other words, we obtained the same target genes in the hybrid regardless of which Ndt80 was tagged for immunoprecipitation. Importantly, the binding positions on the *C. dubliniensis* genome in the hybrid were characteristic of the results in *C. dubliniensis*, specifically 97% of the targets in the hybrid are targets in *C. dubliniensis*, and the positions on the *C. albicans* genome were characteristic of *C. albicans* with 96% of the targets in the hybrid being targets in *C. albicans*. Although we only carried out this experiment with one master regulator, the results independently validate our earlier conclusions based on the much more extensive species-to-species comparisons. These observations confirm our previous conclusion that the extreme differences in regulator-target gene connections observed across these *Candida* species are due to changes in the *cis*-regulatory sequences in the target genes rather than changes in the regulators themselves or differences in the physiological state of the species at the time of analysis.

## Discussion

In this work, we examined how a complex transcriptional network underlying a specific phenotype (*Figure 1*) evolved over a span of approximately 70 million years. The phenotype is biofilm formation by *Candida* species, a group of fungi that colonize humans, sometimes leading to disease. We documented phenotypic differences in biofilm formation across many fungal species and mapped the transcriptional networks underlying biofilm formation in four of them, *C. albicans*, *C. dubliniensis*, *C. tropicalis,* and *C. parapsilosis*. All four species form complex biofilms both *in vitro* and *in vivo* in a rat catheter model (*Figure 2*).

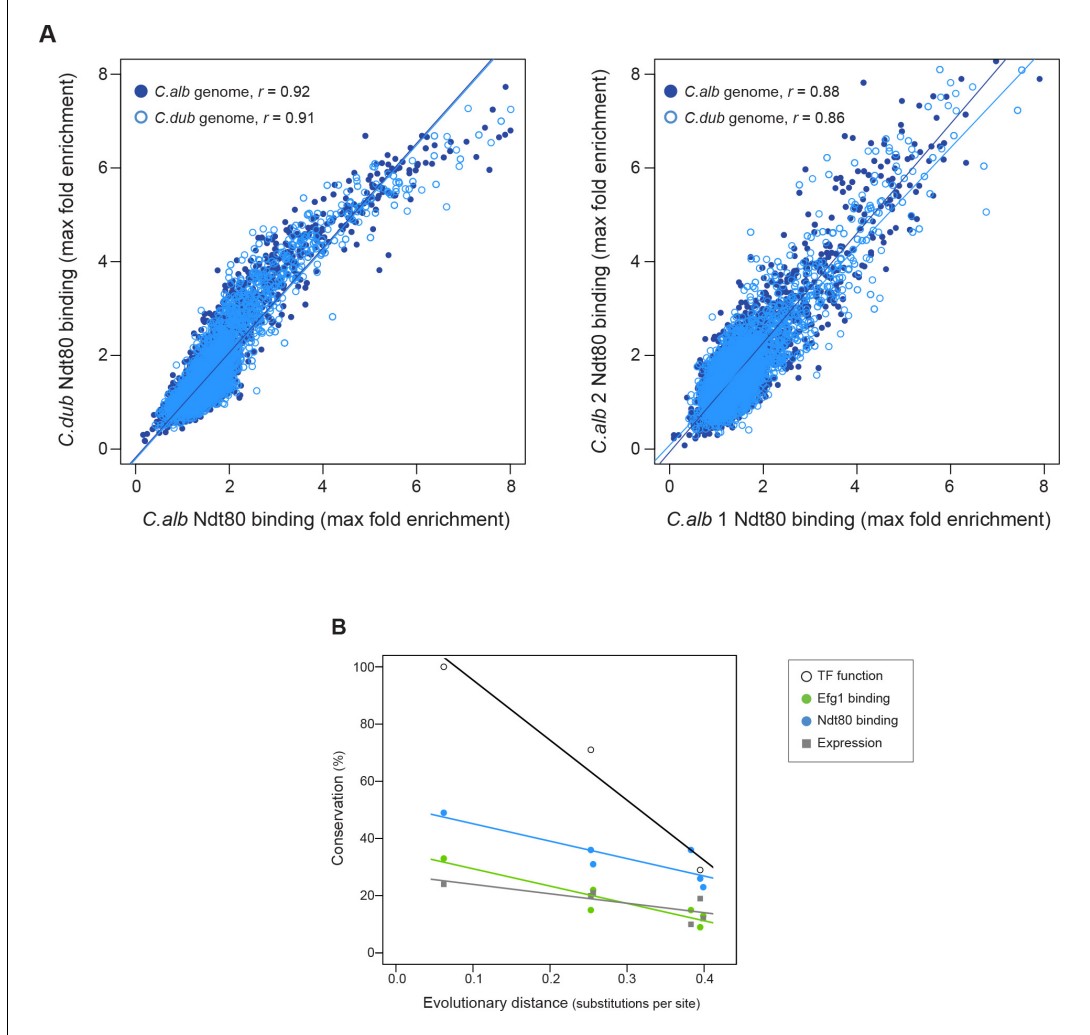

**Figure 6.** Ndt80 ChIP-seq in a hybrid and rate of conservation change of the different network components. (**A**) Genome-wide comparison of *C. albicans* and *C. dubliniensis* Ndt80 binding in the hybrid strain. Binding to both the *C. albicans* (dark blue filled dots) and the *C. dubliniensis* (light blue empty dots) genomes is depicted. The maximal fold enrichment for each upstream intergenic region in the genome is plotted as well as the linear regression for each comparison. The left panel shows the *C. albicans* Ndt80–*C. dubliniensis* Ndt80 comparison while the right panel shows, as a reference, the comparison of the two experimental replicates that are most dissimilar. (**B**) Comparison between the master regulators required for biofilm formation, the Efg1 and Ndt80 binding targets, and biofilm gene expression, as a function of evolutionary distance. Master regulator conservation is depicted as the percentage of *C. albicans* regulators required for biofilm formation. Efg1 and Ndt80 target conservation reflect the percentage of targets shared by the different species pairs. Gene expression conservation represents the number of genes whose expression changes at least 1.5 $\log_2$ fold under biofilm-forming conditions between each species pair. The *C. parapsilosis* gene expression data is from *Holland et al., 2014*, and 1.5 $\log_2$ fold was chosen as a cutoff because this was the cutoff used in this prior study. There are three estimates of master regulator conservation because comparisons were performed between *C. albicans* and each of the other three species, while there are six estimates of binding target and gene expression conservation since comparisons were performed in pairs between all four species. Linear regressions are shown in the corresponding color. Evolutionary distance as substitutions per site was calculated from a phylogenetic tree of these species, inferred from protein sequences of 73 highly conserved genes (*Lohse et al., 2013*).

The online version of this article includes the following figure supplement(s) for figure 6:

**Figure supplement 1.** Morphology of biofilms formed by the *C. albicans*–*C. dubliniensis* tetraploid hybrid (*C. albicans* × *C. dubliniensis*) visualized by confocal scanning laser microscopy.

Using *C. albicans* as a reference species, our analysis leads to the following five main conclusions. (1) As we move away from *C. albicans*, biofilms become less complex, both in terms of structure and of composition; that is, fewer cell types are involved and the resulting biofilm is less regular. At larger evolutionary distances, fungal species did not form biofilms at all under the conditions we

tested. (2) Of the seven master transcriptional regulators of biofilm formation in *C. albicans*, all seven are needed in the most closely related species (*C. dubliniensis*), but, as we move further away evolutionarily, fewer are required for biofilm formation. For example, two of the master regulators (Rob1 and Flo8) are dispensable for biofilm formation in the next most closely related species (*C. tropicalis*), and three of the seven are not required in *C. parapsilosis*. As shown by Holland and colleagues (*Holland et al., 2014*), other transcriptional regulators (present in *C. albicans* but not required for biofilm formation in this species) have assumed the role of master regulators in *C. parapsilosis*. If other master biofilm regulators exist in *C. dubliniensis* and *C. tropicalis* and are identified in the future, their analysis in *C. albicans* and *C. parapsilosis* will allow a more complete model of the biofilm regulatory network across the four species considered here. (3) In contrast to the relatively slow evolutionary substitutions of master regulators, the connections between the master regulators and their target genes have changed very rapidly over evolutionary time (*Figure 4*, *Figure 6B*). This conclusion is most obvious when we compare the two most closely related species, *C. albicans* and *C. dubliniensis*, estimated to have last shared a common ancestor 20 million years ago. Depending on the regulator, fewer than 50% of the master regulator-target gene connections were observed to be conserved (*Figure 4*). This conclusion was independently verified for one regulator — Ndt80 — by analyzing its binding distribution across the two genomes in a *C. albicans*–*C. dubliniensis* hybrid; here, the binding distribution of Ndt80 across one genome differed considerably from that of the other, and each resembled that seen in the cognate individual species (*Figure 6A*). This result strongly supports the conclusion that the differences in regulator-target gene connections across species are due largely to changes in the *cis*-regulatory sequences of the target genes rather than changes in the regulators. (4) As predicted from the extensive changes in regulator-target gene connections, mRNA expression during biofilm formation differs considerably from one species to the next. Like the other changes we have documented in this paper, mRNA expression divergence becomes greater as the phylogenetic distance increases (*Figure 5B*). (5) Despite the extensive changes in the transcription networks underlying biofilm formation across the species we examined, several key features of the overall architecture of the network appear to be preserved. For example, all species show high connectivity in the sense that many target genes are directly connected (by binding) to more than one master regulator. The high connectivity observed is dominated by the DNA binding of Ndt80 and Efg1, and we note that these regulators are also involved in several other cellular functions and have been suggested to cooperatively regulate their target genes (*Sellam et al., 2010*; *Znaidi et al., 2013*; *Mancera et al., 2015*). Moreover, many of the master regulators bind to their own control regions as well as those of the other master regulators. We have argued elsewhere that these two features are likely to be common to many complex transcription networks (*Sorrells and Johnson, 2015*), and the results presented here show that, despite many changes in individual regulator-target connections, the basic 'structural features' of the network are preserved across the biofilm networks of the four species examined.

The large amount of new genome-wide protein-DNA interaction and gene expression data reported here will be useful in future studies of biofilm formation in these *Candida* species. Very few 'structural' genes have been implicated in biofilm formation, and the ChIP-seq and transcriptional profiling results obtained across species could greatly facilitate the identification of key non-regulatory genes required for biofilm formation. For example, specific master regulator-target gene connections that are preserved across multiple species may point to target genes that are notably important for biofilm formation in these *Candida* species. Such a hypothesis could be tested in future studies by deleting these target genes of interest and assessing their roles in biofilm formation across species.

To place our findings in context, it is also instructive to compare our analyses of biofilm formation across species with recent studies where biofilm formation has been analyzed across different isolates of a single species, *C. albicans* (*Hirakawa et al., 2015*; *Huang et al., 2019*). *Hirakawa et al., 2015* determined the genome sequences of 21 clinical isolates of *C. albicans* and examined their abilities to form biofilms. The genome comparisons revealed many differences among strains including aneuploidies, losses of heterozygosity, and mutations in coding sequences; moreover, the strains differed substantially in their abilities to form biofilms. Among the strains analyzed, the one used in our study (SC5314) was among the thickest biofilm producers as assayed by dry weight; the majority of isolates formed thinner biofilms. One clinical isolate that formed very poor biofilms was found to have an inactivating mutation in *EFG1*, one of the biofilm master regulators, indicating a relatively

recent change. Because the *C. dubliniensis* strain used in our study (CD36) formed biofilms that are similar to those of strain SC5314, we believe that SC5314 is a good representative of *C. albicans* and that most of the clinical isolates probably acquired mutations (including aneuploidies and losses of heterozygosity) relatively recently. *Huang et al., 2019* examined five of the previously sequenced strains in much more detail including the dependence on individual transcriptional regulators for biofilm formation. Although the magnitude of the effect of transcriptional regulator deletions on biofilm formation varied across strains, SC5314 again appears to be a good representation of the ability of *C. albicans* as a species to form thick, complex biofilms.

To our knowledge, this study is the first to examine in detail how a complex transcription network changes over a relatively short evolutionary time — 70 million years — represented by four different species. During that time, the master transcription regulators controlling biofilm formation have undergone slow substitutions, but their connections to the target genes they control have changed rapidly. We do not know which, if any, of these changes were adaptive; in this regard, it is important to note that, although the biofilms produced by the *Candida* species have many similarities, they do differ from species to species in at least subtle aspects. Even considering the two most closely related species (*C. albicans* and *C. dubliniensis*), it is possible to distinguish their biofilms. Although they appear very similar under the confocal microscope, the *C. albicans* biofilms form faster and under a greater range of conditions; once formed, they are more difficult to disrupt than those of *C. dubliniensis*. Although these differences may help to explain why *C. albicans* is a greater problem in the clinic than *C. dubliniensis*, it is difficult to reconcile these subtle differences in phenotype with the large differences in the underlying transcriptional circuitry. Given the large magnitude of changes underlying such similar phenotypic output, we propose as a default hypothesis that many of the changes in transcription circuitry result from neutral evolution, more specifically, constructive neutral evolution whereby molecular complexity can change without an increase in fitness (*Stoltzfus, 1999*; *Lynch, 2007*; *Wagner, 2014*; *Sorrells and Johnson, 2015*; *Brunet and Doolittle, 2018*). This study clearly shows that complex transcription networks responsible for the same basic phenotype can undergo evolutionary changes that appear much greater in magnitude than the resulting differences in phenotype.

## Materials and methods

### Characterization of biofilm formation

Visualization of biofilms by CSLM of the different species and strains was performed on silicone squares as described previously (*Nobile et al., 2012*). The strains used for each species and media employed are shown in *Supplementary file 1d and 1a*, respectively. Briefly, for the adhesion phase, silicone squares pretreated with adult bovine serum albumin (BSA) were inoculated to an $OD_{600}$ of 0.5 with cells from an overnight culture grown at 30°C in YPD medium. After incubation for 90 min at 37°C and 200 rpm in the specific medium (*Supplementary file 1a*) for adhesion, the squares were washed with phosphate-buffered saline and then placed in fresh media and incubated for 48 hr at 37°C and 200 rpm. Biofilms of the species that do not grow well at 37°C were grown at 30°C as indicated in *Figure 2*. After 48 hr, the biofilms were stained for 1 hr with 50 mg/mL of concanavalin A-Alexa Fluor 594 conjugate and visualized on a Nikon Eclipse C1si upright spectral imaging confocal microscope using a 40×/0.80W Nikon objective. At least two independent silicone squares were observed per strain analyzed.

Visualization of biofilm formation over time was also performed by CSLM in Spider medium. For each of the four species observed (*C. albicans*, *C. dubliniensis*, *C. tropicalis*, and *C. parapsilosis*), seven independent silicone squares were used for biofilm formation as described above and the biofilms at each square were visualized at 30 min, 4, 8, 12, 24, 48, and 96 hr after the adhesion phase (*Figure 2—figure supplement 1*).

Determination of the biomass dry weight of biofilms of the different species and strains was performed by growing biofilms on the bottoms of 6-well polystyrene plates pretreated with BSA as previously described (*Nobile et al., 2012*). The cells were adhered for 90 min. These assays were performed in a modified Spider medium that contained 1% glucose rather than mannitol as the carbon source. After 48 hr of biofilm formation, supernatants were aspirated and biofilms were scraped and placed to dry on top of a filter paper. Dried biofilms were weighed on an analytic scale

subtracting the weight of a filter paper in which the media without cells was filtered. Five technical replicates were performed per strain as it has been previously done for large screens using this assay (*Nobile et al., 2012*). As was performed for CSLM visualization, strains that do not grow well at 37°C were grown at 30°C.

Biofilm formation in a Bioflux microfluidic device (Fluxion Biosciences) was assayed as described previously (*Gulati et al., 2017*). The medium used was Spider with 1% glucose and without mannitol, and assays were performed at 37 and 30°C for 24 hr.

*In vivo* biofilm formation assays were performed using the rat central-venous catheter infection model as previously described (*Andes et al., 2004*). After 24 hr of infection by the four species tested, biofilm formation on the intraluminal surface of the catheters was observed by scanning electron microscopy. Procedures were approved by the Institutional Animal Care and Use Committee at the University of Wisconsin, Madison (protocol MV1947).

## Generation of gene deletion strains

Gene deletion strains were constructed using a similar fusion PCR strategy as that described by *Noble and Johnson, 2005* employing histidine and leucine auxotrophic strains. Construction of these strains was performed using the *SAT1* flipper strategy as previously described (*Mancera et al., 2019*). All the strains employed and generated in this study are shown in *Supplementary file 1d*. In brief, the two alleles of each regulator in *C. dubliniensis* were subsequently deleted using the *C. albicans HIS1* and *LEU2* genes. In *C. tropicalis,* the first allele was deleted using the *C. albicans LEU2* gene while the second was deleted using the *CaHygB* gene that confers resistance to hygromycin B (*Basso et al., 2010*). To generate the gene deletion cassettes, ~350 bp flanking 5′ and 3′ regions of each regulator were PCR-amplified from genomic DNA and fused to the corresponding auxotrophic/drug resistance marker by fusion PCR. Transformation was performed by electroporation as previously described (*Porman et al., 2011*). Verification of correct integration of the gene deletion cassettes was performed by colony PCR with primers directed to both flanks of the disrupted gene. Final gene deletion confirmation was performed by colony PCR with primers that anneal at the ORF of each regulator. Two independent isolates of each deletion mutant originating from two separate transformations were generated for each regulator deletion. The regulator knockout strains of *C. albicans* and *C. parapsilosis* had been previously generated as part of efforts to generate collections of regulator gene knockout strains (*Homann et al., 2009*; *Holland et al., 2014*).

## Chromatin immunoprecipitation followed by sequencing

ChIP-seq to identify the target genes of the seven regulators was performed as previously described (*Hernday et al., 2010*; *Lohse and Johnson, 2016*) and sequenced using Illumina HiSeq 2500 or 4000 platforms. Each of the seven regulators in *C. dubliniensis* and *C. tropicalis* was tagged in the wildtype strain background with a 13× Myc epitope tag at the C-terminus from the pADH34 or pEM019 plasmids, respectively, as previously described (*Hernday et al., 2010*; *Mancera et al., 2019*). *C. albicans* Myc-tagged strains had been similarly generated previously (*Nobile et al., 2012*). *C. parapsilosis* Brg1, Ndt80, and Tec1 were tagged using a 6× C-terminal Myc tag amplified from plasmid pFA-MYC-HIS1 as previously described (*Connolly et al., 2013*). *C. parapsilosis* Efg1 had been previously tagged with the same 6× C-terminal Myc epitope (*Connolly et al., 2013*). We were not able to tag the regulator Bcr1 in this species. Genotype details for all the strains generated and used are given in *Supplementary file 1d*. All the tagged regulator strains were tested for their abilities to form biofilms on the bottoms of 6-well polystyrene plates as previously described (*Nobile et al., 2012*), and no biofilm defects were observed.

To generate the *C. albicans*/*C. dubliniensis*-tagged hybrid strains, the α or **a** allele of the mating-type-like (*MTL*) locus was deleted in the Ndt80-tagged strains described above. These deletions allowed the strains to become capable of white-opaque switching, and thus mating competent. In *C. albicans,* the α allele of the *MTL* locus was deleted by replacing it with *ARG4* using the plasmid pJD1 as previously described (*Lohse et al., 2016*). In *C. dubliniensis,* the **a** allele of the *MTL* locus was deleted using a cassette containing the *SAT1* nourseothricin resistance marker from plasmid pSFS2A flanked by ~300 bp homology regions identical to the 3′ and 5′ upstream/downstream regions of the *MTL* locus. pSFS2A is a plasmid derived from pSFS2 (*Reuss et al., 2004*) that contains the *SAT1* reusable cassette in the backbone of vector pBC SK+ instead of pBluescript II KS and that

was kindly provided by Joachim Morschhauser (U. Würzburg). The deletion of the α or **a** alleles was verified by colony PCR of the two flanks. Generation of the hybrid strains was done by overlaying the mating competent wildtype and Myc-tagged strains on a YPD plate for 48 hr at 30℃. Single colonies were then streaked out and hybrids were selected by growing on media containing nourseothricin and lacking arginine. The hybrid strains were further verified measuring DNA content using FACS. As controls, the two wildtype untagged mating competent strains were hybridized.

All immunoprecipitation experiments were performed under biofilm growth conditions in 6-well polystyrene plates as previously described (*Nobile et al., 2012*). After 48 hr of biofilm growth in Spider 1% glucose at 37℃ and 200 rpm, cells were fixed with 1% formaldehyde for 15 min. Cell disruption and immunoprecipitation were performed as previously described (*Hernday et al., 2010*) using a c-Myc tag monoclonal antibody (RRID:AB_2536303). After crosslink reversal, instead of performing a phenol/chloroform extraction, we used a MiniElute QIAGEN kit to purify the immunoprecipitated DNA. Library preparation for Illumina sequencing was performed using an NEBNext ChIP-Seq Library Prep Master Mix Set for Illumina sequencing. Between 12 and 24 samples were multiplexed per lane. As controls, immunoprecipitations were performed in matched strains that lacked the Myc tag. In agreement with the ChIP-seq guidelines and practices of the ENCODE consortia (*Landt et al., 2012*), two biological replicates were performed for each regulator in the four species.

## Identification of regulator directly bound target genes by ChIP-seq

ChIP-seq reads were mapped to their corresponding genome using Bowtie 2 with default parameters (*Langmead and Salzberg, 2012*). The genome sequences and annotations were obtained from CGD (*Skrzypek et al., 2017*) for versions: C_albicans_SC5314_version_A21-s02-m09-r02, C_dubliniensis_CD36_version_s01-m02-r08, C_tropicalis_MYA-3404_2013_12_11, and C_parapsilosis_CDC317_version_s01-m03-r13. The SAMtools package was used to convert, sort, and index the sequenced reads to BAM format (*Li et al., 2009*). We observed that the peak calling algorithm was more specific and sensitive if the number of reads in the treatment and control datasets was similar. Therefore, we adjusted the number of reads in the different treatment-control dataset pairs using SAMtools view -s function prior to peak calling. Peak calling was performed using MACS2 (*Zhang et al., 2008*) with a q-value cutoff of 0.01; the shiftsize parameter was determined using the SPP package in R (*Kharchenko et al., 2008*). Peaks were considered as true binding events only if the peak was identified in both biological replicates. Assignment of peaks to ORFs was done using MochiView when the peak was present in the intergenic region immediately upstream of the ORF (*Homann and Johnson, 2010*).

To identify regulator binding target genes in the hybrid strains, ChIP-seq reads were aligned to the *C. albicans* and *C. dubliniensis* genomes as described above. Reads that aligned to both genomes were subsequently filtered out. Further processing, peak calling, and assignment of peaks to ORFs were then performed independently for reads that mapped to the *C. albicans* and *C. dubliniensis* genomes as described above.

## *De novo* sequence motif discovery and enrichment for the regulators

DNA-binding motifs were generated *de novo* for regulators from ChIP-Seq experiments using DREME (*Bailey, 2011*). The union of the sequences under the peaks of the two biological replicates for each experiment was tested against a background of equivalent length random genomic sequences from that species. The top-scoring motif was taken and is shown in *Figure 5*.

Based on the motifs generated *de novo* using DREME as well as previously reported DNA-binding motifs (*Lassak et al., 2011*; *Nobile et al., 2012*; *Connolly et al., 2013*; *Nocedal et al., 2017*), a high-confidence 'consensus motif' was generated for Ndt80 (CACAAA) and Efg1 (TGCAT). To determine enrichment of these consensus motifs in peaks identified for each ChIP-seq experiment, the number of consensus motifs in the union of the sequence under the peaks of both biological replicates was compared to the number of motifs in intergenic regions for that species. A Fisher's one-tailed exact test was performed to generate a p-value representing enrichment of the motif in peaks compared to equivalent length random intergenic sequences.

To determine potential gene targets based on motif presence, we scored the presence of the *de novo* DNA-binding motifs of Efg1 or Ndt80 generated for each species described above in all the intergenic regions of the four species using the Motif scoring function of MochiView (*Homann and*

*Johnson, 2010*). Potential gene targets were defined as those having at least one motif in their upstream intergenic region. Then, for each species, the overlap was calculated between the potential gene targets found using the motifs derived from its own ChIP-seq data and the potential gene targets found using the motifs derived from the ChIP-seq data of each of the other three species.

## Genome-wide transcription profiling

Cultures for the extraction of total RNA under biofilm growth conditions were performed on biofilms grown on the bottom of 6-well polystyrene plates for 48 hr at 37°C and 200 rpm as previously described for the determination of biofilm biomass dry weight (*Nobile et al., 2012*). The media used was Spider 1% glucose for all species, Spider for *C. albicans* and *C. dubliniensis*, and RPMI 1% glucose for *C. albicans* and *C. tropicalis*. Planktonic cultures for total RNA were grown in the corresponding media by inoculating with cells from an overnight 30°C YPD culture to an $OD_{600}$ of 0.05. Cultures were then grown in flasks at 37°C with shaking at 225 rpm until they reached an $OD_{600}$ of 1.0. Biofilm and planktonic cultures were harvested immediately by centrifugation at 3000 g for 3 min and snap-frozen in liquid nitrogen. Total RNA was extracted from the frozen pellets using the RiboPure-Yeast RNA kit (Ambion, AM1926) following the manufacturer's recommendations. Transcription profiling was performed by hybridization to custom-designed Agilent 8*15 k oligonucleotide microarrays that contain between 2 and 3 independent probes for each ORF (*C. albicans* AMADID #020166; *C. dubliniensis* AMADID #042592; *C. tropicalis* AMADID #042593). cDNA synthesis, dye coupling, hybridization, and microarray analysis was performed as previously described (*Nobile et al., 2012*). In agreement with previous reports (*Nobile et al., 2012*; *Nocedal et al., 2017*), two biological replicates were performed for each species in each condition using the wild-type strains. The genes that are differentially expressed in *C. parapsilosis* during biofilm formation were obtained from *Holland et al., 2014*, Table_S3.xls, where a cutoff of 1.5 $\log_2$ fold change was used to define differentially expressed genes.

## Data deposition

ChIP-seq and microarray gene expression data has been deposited to the NCBI Gene Expression Omnibus (GEO) repository under Superseries GSE160783.

# Acknowledgements

We thank Derek Sullivan and especially Joachim Morschhäuser for providing strains, plasmids, and advice, and Victor Hanson-Smith for help with the ChIP-seq data analysis.

# Additional information

### Competing interests

Clarissa J Nobile, Alexander D Johnson: cofounder of BioSynesis, Inc, a company developing diagnostics and therapeutics for biofilm formation. The other authors declare that no competing interests exist.

### Funding

| Funder | Grant reference number | Author |
|---|---|---|
| Human Frontier Science Program | LT000484/2012-L | Eugenio Mancera |
| UC MEXUS | | Eugenio Mancera |
| Consejo Nacional de Ciencia y Tecnología | CB-2016-01 282511 | Eugenio Mancera |
| Wellcome Trust | 209077/Z/17/Z | Eugenio Mancera |
| National Institutes of Health | R01AI083311 | Alexander D Johnson |
| National Institutes of Health | R01AI049187 | Alexander D Johnson |

| National Institutes of Health | R01AI073289 | David R Andes |
| National Institutes of Health | R35GM124594 | Clarissa J Nobile |
| National Institutes of Health | R21AI125801 | Clarissa J Nobile |
| Pew Charitable Trusts | Pew Biomedical Scholar Award | Clarissa J Nobile |
| Kamangar family endowed chair | | Clarissa J Nobile |

The funders had no role in study design, data collection and interpretation, or the decision to submit the work for publication.

## Author contributions
Eugenio Mancera, Conceptualization, Formal analysis, Supervision, Funding acquisition, Investigation, Writing - original draft, Writing - review and editing; Isabel Nocedal, Formal analysis, Investigation, Writing - review and editing; Stephen Hammel, Megha Gulati, Kaitlin F Mitchell, Investigation; David R Andes, Supervision, Funding acquisition; Clarissa J Nobile, Geraldine Butler, Supervision, Funding acquisition, Writing - review and editing; Alexander D Johnson, Conceptualization, Supervision, Funding acquisition, Writing - original draft, Writing - review and editing

## Author ORCIDs
Eugenio Mancera (iD) https://orcid.org/0000-0003-0146-8732
Isabel Nocedal (iD) http://orcid.org/0000-0002-4706-1113
Clarissa J Nobile (iD) https://orcid.org/0000-0003-0799-6499
Geraldine Butler (iD) http://orcid.org/0000-0002-1770-5301

## Ethics
Animal experimentation: Procedures were approved by the Institutional Animal Care and Use Committee (IACUC) at the University of Wisconsin, Madison (protocol MV1947).

## Decision letter and Author response
Decision letter https://doi.org/10.7554/eLife.64682.sa1
Author response https://doi.org/10.7554/eLife.64682.sa2

# Additional files

## Supplementary files
• Supplementary file 1. Supplementary Tables. (**a**) Biofilm formation of *C. albicans* and its three most closely related species in different media. Biofilms were grown on silicone squares at 37°C with shaking at 200 rpm. Biofilm formation was assessed by confocal scanning laser microscopy after 48 hr of growth. Spider medium was prepared with 1% nutrient broth, 0.4% potassium phosphate, and adjusted to pH 7.2. RPMI media used was RMPI 1640 with 165 mM MOPS and L-glutamine and without sodium bicarbonate (Lonza, 04-525F). YNB media was prepared with 0.67% yeast nitrogen base with ammonium sulfate. Question marks denote conditions where it was not possible to visualize the biofilms since the dye did not stain them. (**b**) Number of genes bound by each regulator in the four *Candida* species studied. 'NA' indicates experiments that were not performed because the regulator was not tagged (*C. parapsilosis* Bcr1 and Flo8) or due to the absence of an ortholog (*C. parapsilosis* Rob1). Zeros denote experiments for which the ChIP-seq was not successful. (**c**) Master regulator binding to each other's upstream intergenic regions. Numbers represent the number of species in which binding was observed. The maximum possible number of species for which we have data is shown in parenthesis for each master regulator. (**d**) Strains used in this study.

• Transparent reporting form

## Data availability

ChIP-seq and microarray gene expression data has been deposited to the NCBI Gene Expression Omnibus (GEO) repository under Superseries GSE160783.

The following dataset was generated:

| Author(s) | Year | Dataset title | Dataset URL | Database and Identifier |
|---|---|---|---|---|
| Mancera E, Nocedal I, Hammel S, Gulati M, Mitchell KF, Andes DR, Nobile CJ, Butler G, Johnson AD | 2021 | Evolution of biofilm formation in Candida | https://www.ncbi.nlm.nih.gov/geo/query/acc.cgi?acc=GSE160783 | NCBI Gene Expression Omnibus, GSE160783 |

The following previously published dataset was used:

| Author(s) | Year | Dataset title | Dataset URL | Database and Identifier |
|---|---|---|---|---|
| Holland LM, Schröder MS, Turner SA, Taff H, Andes D, Grozer Z, Gacser A, Ames L, Haynes K, Higgins DG, Butler G | 2014 | Comparative phenotypic analysis of the major fungal pathogens Candida parapsilosis and Candida albicans | https://www.ncbi.nlm.nih.gov/geo/query/acc.cgi?acc=GSE57451 | NCBI Gene Expression Omnibus, GSE57451 |

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
