## [Decision Letter]

**Acceptance summary:**

Mancera and colleagues examine the evolution of the regulatory circuitry involved in biofilm formation in *Candida albicans* and closely related species. Using a combination of genomics approaches that they apply in a comparative manner across closely related species, they show that some features of this network have changed significantly, while others are conserved. This work contributes to a better understanding of how traits that are key to the success of opportunistic pathogenic fungi evolve through changes in regulatory networks.

**Decision letter after peer review:**

Thank you for submitting your article "Evolution of the complex transcription network controlling biofilm formation in *Candida* species" for consideration by *eLife*. Your article has been reviewed by 3 peer reviewers, and the evaluation has been overseen by a Reviewing Editor and Patricia Wittkopp as the Senior Editor. The following individuals involved in review of your submission have agreed to reveal their identity: Bin He (Reviewer #1); Mira Edgerton (Reviewer #2); Sadri Znaidi (Reviewer #3).

The reviewers have discussed the reviews with one another and the Reviewing Editor has drafted this decision to help you prepare a revised submission.

Summary:

Mancera and colleagues examine the evolution of the regulatory circuitry involved in biofilm formation in Candida and other closely related species. Using a combination of genomics approaches that they apply in a comparative manner across closely related species, they show that this network has changed significantly while some features have been conserved.

Three experts with complementary expertise have evaluated your manuscript. They all identified key issues that would need to be addressed in a revised version of the manuscript. I added the list of comments below.

Essential revisions:

1. The first reviewer suggests many approaches that would help anchor the results and analysis in an evolutionary framework. Some of them are critical to the paper, for instance considering the species non-independence when performing comparative analysis. Other issues related to statistical analysis are also critical. The changes requested will significantly strengthen the quality and impact of your work.

2. One reviewer was more critical about the inclusion of more distantly related species because the conditions associated with biofilm formation differ for those. However, another reviewer found these experiments important. I suggest you keep them in the paper but include a discussion of these points in the manuscript.

3. The third reviewer suggests ways in which the text can be clarified and made easier to follow.

Reviewer #1:

This work produced highly valuable phenotypic and molecular binding datasets for an important phenotype, namely biofilm formation, which will have significant impact both for the medical community in terms of better understanding and potentially advancing the treatment for candida infections, and also in the evolution of complex transcriptional networks. The major weakness of the manuscript as noted in the detailed review below is that it didn't connect the molecular binding differences with the differences in gene expression and phenotype, and some of the statistical tests didn't properly account for the phylogenetic relationships between the species and thus are likely invalid.

In this work Mancera and co-authors built upon a series of significant findings from the Johnson lab on the architecture and evolution of the biofilm network, and produced highly valuable datasets for future research. The key question in this work is how a complex network such as the one controlling biofilm formation could evolve over a relatively short amount of time. To do so, they focused on two closely related species of *C. albicans* to determine the evolution of the key biofilm regulators and their targets. By performing detailed phenotypic comparisons and collecting large amount of functional genomics data, they reached five main conclusions, including 1) enhanced and more robust biofilm forming abilities in *C. albicans*, followed by its close relatives, 2) the slow substitution of the regulators, their DNA specificity and interacting partners, and yet fast divergence in their targets. The latter, namely trans factors evolved more slowly than their cis counterparts is consistent with previous studies in other systems, such as *Drosophila* species and mammals (Wilson et al. 2008, the human chromosome 21 placed in a mouse cell). This paper made several important contributions. First, it performed careful phenotypic assays to characterize the biofilm phenotypes under different inducing conditions in a group of closely related and medically relevant species. Second, it provided a rich set of molecular binding and gene expression data for future studies to identify and test the importance of individual regulators and their targets' role in biofilm.

While reading the manuscript, I had a number of concerns and what I see as missed opportunities for the authors to realize the full potential of their work. First and foremost, this work produced three categories of comparative data and yet failed to connect them. These three data categories are phenotypic (biofilm characteristics), gene expression (RNA-seq) and regulators binding (ChIP). An obvious causal relationship among the three categories exists, namely divergence in regulator binding leading to changes in gene expression, which in turn results in phenotypic divergence. However, in the most part the authors analyzed and presented the data as separate entities, not allowing an examination on their causal relationships. I acknowledge that it is difficult to make explicit correlations between the three given the complexity of the network. However, certain analyses should be attempted and seem feasible. For example, one could analyze the extent to which the observed differences in regulator binding correlate with the divergence in gene expression during biofilm formation. Without such analyses, the three categories of data are disconnected and don't allow for a deep understanding of how the complex network evolved.

A related issue is that the results are primarily descriptive, without inferences about the ancestral state and the direction of evolution. This is particularly true of the binding data, where the major types of analyses were about overlaps between the species. This feeds into the issue above as it is not clear how those differences in binding relate to the differences in the biofilm phenotype. Notably, the authors used the words "evolve" and "diversity" in several context, which to me imply a direction in evolution, while the presented analyses only concern differences between species.

A second concern is that in most of the analyses the authors considered a gene as a "target in the biofilm network" if orthologs of one of the seven major regulators of biofilm formation in *C. albicans* bound in its upstream region (as evidenced by ChIP). It is well known that TF binding is not always productive, that is, leading to gene induction or repression, which in turn don't always affect the phenotype. This raises a serious question in my mind: how much of the observed binding difference actually relates to the biofilm phenotype vs stochastic changes due to neutral evolution? I could imagine a model in which only a small percentage of the binding events are functional, while the rest represent spurious binding motifs that evolve more or less neutrally. The observed difference could then be due to genetic drift in the gain and loss of short DNA motifs. The authors did include a few additional criteria in their comparative analyses, including the requirement for a high scoring motif in the binding region and a two-fold change in the gene's expression under biofilm forming conditions. These do not completely address the concern above and were also presented towards the end of the Results section, leaving me as a reader pondering on the above question for most of the manuscript. In my opinion, the issue of binding ≠ function should be addressed head-on and controlled for using complementary data, such as RNA-seq of the wild type and TF-KO strain under biofilm formation conditions. If that's not possible, the authors should attempt to estimate the proportion of ChIP identified sites as being truly in the biofilm network.

Third, I have doubts about the statistical tests behind a number of conclusions in the paper, e.g. on lines 337-340, the authors said "the overlap for each species pair is larger than expected by chance (hypergeometric test, P < 0.05), indicating that a small but significant group of Ntd80-target gene connection have been preserved across these species", and on lines 407 to 410, the authors said "the number of connections observed between the regulators is greater than what would be expected by chance (P << 0.05).… high connectivity between the master regulators is conserved in the four *Candida* species analyzed". Based on the description, I think the authors performed the tests under the null hypothesis that the regulator-target relationships evolved independently in these species, while they were actually non-independent due to the species sharing a common ancestor. The concept of phylogenetically independent contrast and a number of statistical tests based on it was developed so as to take into account the relatedness among the species being compared. One would need to specify a neutral rate at which a trait evolves, and compare the observed level of divergence to the expected level in order to make conclusions about their conservation due to selective constraint.

Reviewer #2:

This comprehensive study describes the relatedness and evolution of biofilm specific genes among several Candida strains. The strength and impact of this study is elucidating the transcriptional regulators controlling biofilms with resulting functional differences in biofilm formation among closely and distantly related *Candida* species.

*Candida* species differ in their relative ability to form biofilms, and this study documents differences in the transcriptional networks among species that influence this phenotype. This work shows that among seven master transcriptional regulators needed for robust *C. albicans* biofilm formation, fewer are involved in biofilm formation in more distantly related species and this accompanies less organized or poorer biofilm production. The authors find that this network of regulator-target connections are predictive of evolutionary changes across species and provide insight into key components involved in biofilm formation.

This manuscript comprises an impressive breadth of experiments, yet the data is easy to follow and the figures are understandable synthesis of an enormous amount of data. The biofilm conditions selected are comprehensive and resulting gene expression data well validated and statistically supported. There are major formatting errors in Suppl Figure 4 , and minor errors in Suppl Figure 1 (top legend) and Suppl Figure 2 ( species names below blank ?).

Reviewer #3:

This work highlights similarities and differences in the way pathogenic *Candida* species control biofilm formation, contributing to our understanding of how pathogenic traits evolved in species of the Candida lineage. Although this study is limited to four related species, with some overstated conclusions, it could serve as a nice example for how transcriptional circuits operate in microbial pathogens that recently shared a common ancestor.

An important virulence trait of fungal pathogens is their ability to form biofilms, which are communities of adherent cells encased in a polymeric matrix acting as protective structures. Transcriptional control of biofilm development is a complex process, involving a plethora of regulators with variable impact on both structure and function of biofilms. Here authors build on their previous findings to further map the transcriptional regulatory network that controls biofilm formation in four pathogenic *Candida* species of medical importance, namely *C. albicans*, *C. dubliniensis*, *C. tropicalis* and *C. parapsilosis*. They provide interesting clues as to how such a circuitry operates in these species. Although authors attempted to include some additional species from the Candida and Saccharomyces lineages in their analyses, they are faced with the complexity of the environmental cues/specificities that trigger biofilm formation in these more distantly-related organisms, reflecting the challenge of studying and explaining the evolution of some traits over extended evolutionary time scales. This study could've been certainly more impactful and comprehensive if some technical issues had been addressed/resolved (e.g. failure of tagging/expressing a subset of transcription factors in the species being analyzed/compared) or if some conclusions had been tempered. Collectively, authors present a valuable work on the transcriptional control of an important pathogenicity trait in medically-important *Candida* species and extend the reach of the regulatory circuitry operating during biofilm formation in fungal pathogens.

In this work, Mancera and colleagues investigate the transcriptional circuitries that control biofilm development in four medically-important *Candida* species using functional genomics. They characterize a subset of regulators that were already studied in *Candida albicans* and *Candida parapsilosis* in major publications by the groups involved in this study (Nobile et al. 2012, Holland et al. 2014) in two additional related species from the Candida lineage, namely *Candida tropicalis* and *Candida dubliniensis*. Although authors attempted to include some additional species from the Candida and Saccharomyces lineages in their analyses, they are faced with the complexity of the environmental cues/specificities that trigger biofilm formation in these more distantly-related organisms, reflecting the challenge of studying and explaining the evolution of some traits over extended evolutionary time scales. Still, authors provide interesting clues as to how the transcriptional circuitry that controls biofilm development operates in the four phylogenetically-close species being compared. This study could have been more impactful and comprehensive if some technical issues had been addressed/resolved (e.g. failure of tagging/expressing a subset of transcription factors and getting ChIP-seq data in the species being analyzed/compared) or if some non-useful data (e.g. other CTG species, *S. cerevisiae*/*C. glabrata*) had been removed and conclusions tempered (see some examples from specific comments).

Specific comments:

– Authors should rather focus on biofilm development in the 4 *Candida* species, because conditions have been optimized for the set of 4 species. For those distantly-related species, the conditions and biofilm development might not be similar at all. For instance, *S. cerevisiae* does make biofilms but in quite different ways and under different conditions/stimuli (e.g.#1, mat formation, e.g.#2, glucose inhibits adhesion and biofilm formation in *S. cerevisiae*, see PMID: 32054862).

– Lines 189-207, description of results but no data are shown. This section does not bring much information and the conclusions are overstated (e.g. "the greater the phylogenetic distance from *C. albicans*, the thinner the biofilm formed") because in distantly-related species the conditions conducive to biofilm formation may not be similar to those optimized for *C. albicans*, *C. dubliniensis*, *C. tropicalis* and *C. parapsilosis*. Same for lines 210-217.

– Lines 234-236: "Of all the species studied (…) to physical manipulation (results not shown)". This is again an overstatement. Authors optimized biofilm conditions based on experiments performed on only 4 closely-related species.

– Figure 2: *C. tropicalis* CSLM data are missing in Panel B.

– Figure 3, Supp Figure 3 and lines 270-271: "All seven master regulators identified in *C. albicans* were also required for biofilm formation in *C. dubliniensis*" – However this does not appear to stand true for *C. dubliniensis* BRG1. Any explanations?

– Figure 3 and lines 271-284: Data from *C. albicans* and C. dublinienesis should also be shown in Figure 3 to serve as a reference/benchmark. The conclusion pertaining to this figure still needs to be tempered (i.e. lines 286-287, "with the degree of diversity roughly paralleling their evolutionary distance from *C. albicans*"). No clear evidence supports such a conclusion, because observations were made based on mutant phenotypes from only 4 closely-related *Candida* species.

– Lines 304-309: This is a major weakness in this manuscript. First, did author test the functionality of the tagged transcription factors (TFs)? What about their expression level in biofilm-stimulating conditions? The fact that authors failed to ChIP some of them might be due to epitope tagging which could have altered their function. Which TFs failed to be tested? It is not clearly stated in the manuscript. Again, failure to clearly show which regulators have been successfully ChIPed and the lack of data from those regulators weakens the manuscript. Maybe authors should give them a second try (by tagging differently C-term vs N-term/testing the functionality?)

– Supp Table 2 is a key table and should be included in the main text. Legend is missing, it is not clear what "N/A" vs "0" stand for? It appears that Ndt80 and Efg1 targets account for the majority of binding events in the species being studied. Unlike the other regulators, these are major regulators for many important processes in *Candida* species including, for instance, filamentous growth. Consequently, the high connectivity of the biofilm network would not be surprising if one takes the Ndt80/Efg1 network as an example. This has been also shown for the Sfl1/Sfl2 transcriptional circuitry that controls filamentous growth in *C. albicans* (PMID: 23966855).

– Figure 4B/C: It is not advised to show only percentages, as we could have the impression that we are dealing with big numbers whereas in reality we are not (e.g. Rob1, only 20 targets). Same for lines 340-343: the low number of Rob1 targets should prevent authors from drawing strong conclusions.

– Lines 345-355 and 369-385: These sections should rather go to the Discussion section or should be rewritten as data originating from experiments (i.e. results per se, not discussion). In many occurrences, authors discuss their data in the Results section. This strongly alters the quality of the reading flow. Same for lines 404-408, appearing as data but could also be moved to the Discussion section.

– Lines 415-441 and Supp Figure 4: This section is rather technical in nature and should have been presented (or summarized) earlier in the manuscript (may be following the ChIP-seq section, lines 290-343). Still, Ndt80 is not a good example for performing robustness analyses with regard to the specific conditions under which biofilm formation was induced, because this major regulator appears to exert pleiotropic functions.

[Editors' note: further revisions were suggested prior to acceptance, as described below.]

Thank you for resubmitting your work entitled "Evolution of the complex transcription network controlling biofilm formation in *Candida* species" for further consideration by *eLife*. Your revised article has been evaluated by Patricia Wittkopp (Senior Editor) and a Reviewing Editor.

The manuscript has been improved but there are some remaining issues identified by one of the two reviewers that need to be addressed, as outlined below:

1. Regarding the new text added on page 21-22, lines 506-570 (there is a jump from 506 to 560), I understand that the goal is to determine whether trans-changes, specifically the DNA binding specificity, could explain part of the observed binding target divergence. However, I can't quite follow the text, as it is not clear to me which species' motif was used to predict in which species' genome, and how were the overlaps actually calculated. I did look for additional information in the Methods section and couldn't find any.

2. Regarding the new text from lines 619-632: am I correct in that the new results were meant to determine the relationship between binding and gene induction *in each species*, rather than attributing gene induction *differences between species* to differences in TF binding in those species? The reason for the question is because I came in expecting answers to the latter question and got confused for a moment.

3. In lines 646-649, the authors laid out the challenges involved in between-species comparison, which I fully agree with. But I don't think the Ca-Cd hybrid experiment can address that. It does address a different question, i.e. specifically revealing cis-changes behind gene expression divergence, which I feel is different from the first one.

4. In Figure 6A, I wonder if the authors can comment on the concave shape of the point cloud on the left.

5. In Figure 6B, there are more than one data points for the two binding and one expression change datasets in the middle and right time points. Are those biological replicates?

6. In the Discussion section, the authors stated that this work examined how a complex transcriptional network underlying a specific phenotype (biofilm formation) evolved over a span of ~70 million years. I think it would be useful to point out, the authors did this to some extent later, that to fully reconstruct the evolutionary history of this network, it is critical to identify all the regulators in the other three species, and that the data in this work constitutes a partial picture for the three non-albicans species.

---

## [Author Response]

Essential revisions:Reviewer #1:[…]While reading the manuscript, I had a number of concerns and what I see as missed opportunities for the authors to realize the full potential of their work. First and foremost, this work produced three categories of comparative data and yet failed to connect them. These three data categories are phenotypic (biofilm characteristics), gene expression (RNA-seq) and regulators binding (ChIP). An obvious causal relationship among the three categories exists, namely divergence in regulator binding leading to changes in gene expression, which in turn results in phenotypic divergence. However, in the most part the authors analyzed and presented the data as separate entities, not allowing an examination on their causal relationships. I acknowledge that it is difficult to make explicit correlations between the three given the complexity of the network. However, certain analyses should be attempted and seem feasible. For example, one could analyze the extent to which the observed differences in regulator binding correlate with the divergence in gene expression during biofilm formation. Without such analyses, the three categories of data are disconnected and don't allow for a deep understanding of how the complex network evolved.

As suggested by the reviewer, we have included an analysis of the overlap between the DNA binding data of the regulators and the gene expression of their target genes (ln. 619). Additional analyses will require extensive experimentation, especially given the number of regulators and species considered in the current study; we have plans to carry such experiments in future studies, where their findings can be published in detail.

A related issue is that the results are primarily descriptive, without inferences about the ancestral state and the direction of evolution. This is particularly true of the binding data, where the major types of analyses were about overlaps between the species. This feeds into the issue above as it is not clear how those differences in binding relate to the differences in the biofilm phenotype. Notably, the authors used the words "evolve" and "diversity" in several context, which to me imply a direction in evolution, while the presented analyses only concern differences between species.

We think it is not currently possible to infer accurately a specific ancestral state from the type of data presented in this paper. In Tuch et al. 2008 (PLoS Biology 6(2):e38) we inferred, using a maximum likelihood approach, the size of an ancestral circuit, but there was no credible way to accurately infer the details of the circuit. In the case of this work it is clear that an ancestral circuit existed and that it changed over time to give rise to the extant species we analyzed. Although we cannot provide specific details, there is no doubt that evolution has occurred, with the “direction” being ancestral to modern.

A second concern is that in most of the analyses the authors considered a gene as a "target in the biofilm network" if orthologs of one of the seven major regulators of biofilm formation in C. albicans bound in its upstream region (as evidenced by ChIP). It is well known that TF binding is not always productive, that is, leading to gene induction or repression, which in turn don't always affect the phenotype. This raises a serious question in my mind: how much of the observed binding difference actually relates to the biofilm phenotype vs stochastic changes due to neutral evolution? I could imagine a model in which only a small percentage of the binding events are functional, while the rest represent spurious binding motifs that evolve more or less neutrally. The observed difference could then be due to genetic drift in the gain and loss of short DNA motifs. The authors did include a few additional criteria in their comparative analyses, including the requirement for a high scoring motif in the binding region and a two-fold change in the gene's expression under biofilm forming conditions. These do not completely address the concern above and were also presented towards the end of the Results section, leaving me as a reader pondering on the above question for most of the manuscript. In my opinion, the issue of binding ≠ function should be addressed head-on and controlled for using complementary data, such as RNA-seq of the wild type and TF-KO strain under biofilm formation conditions. If that's not possible, the authors should attempt to estimate the proportion of ChIP identified sites as being truly in the biofilm network.

As also suggested by reviewer 3, we have moved the section where we integrated different criteria to determine gene targets after we first present the results about the target gene connections so that the issue is addressed early on in the manuscript (ln. 403).

In our experience, determining whether the binding of a regulator is functional is one of the most challenging aspects of analyzing a transcriptional network in any species. As mentioned above, given the number of regulators and species considered in this study, experiments to rigorously establish this point for each piece of binding data are currently beyond what is feasible in our laboratories. We believe that the integration of motif presence and gene expression change during biofilm formation, combined with the binding data, give an adequate estimation of the target gene connections that are most important for biofilm formation in the conditions assessed.

In addition, even with the incorporation of additional experimental data, it is not always possible to show that a binding site is functional; for example, it could only be functional under very specific conditions. Therefore, its modification would only result in a measurable phenotype in those conditions. Given that it is not possible to evaluate every possible condition, there will always be some uncertainty regarding the functionality of binding events in genome-wide binding studies.

Third, I have doubts about the statistical tests behind a number of conclusions in the paper, e.g. on lines 337-340, the authors said "the overlap for each species pair is larger than expected by chance (hypergeometric test, P < 0.05), indicating that a small but significant group of Ntd80-target gene connection have been preserved across these species", and on lines 407 to 410, the authors said "the number of connections observed between the regulators is greater than what would be expected by chance (P << 0.05).… high connectivity between the master regulators is conserved in the four *Candida* species analyzed". Based on the description, I think the authors performed the tests under the null hypothesis that the regulator-target relationships evolved independently in these species, while they were actually non-independent due to the species sharing a common ancestor. The concept of phylogenetically independent contrast and a number of statistical tests based on it was developed so as to take into account the relatedness among the species being compared. One would need to specify a neutral rate at which a trait evolves, and compare the observed level of divergence to the expected level in order to make conclusions about their conservation due to selective constraint.

This is an important point that is worth clarifying in the manuscript. The aim of these statistical tests was not to evaluate whether the binding connections are evolving under a selective constraint. Although we would be very interested in testing this, with the available data, we do not think this is feasible; there is simply not enough information to credibly estimate a neutral rate at which binding connections are evolving to see whether the rate of change that we are observing deviates from it. Instead, these tests were aimed at assessing whether the binding connections have changed at such a fast rate that there is no evidence of common decent anymore, independently of whether they are evolving by natural selection or drift. We have now clarified this point in lines 337-340 (now ln. 360) and also when we discuss the overlap between differentially expressed genes (ln. 613).

In Figure 6B we compared how the different datasets that we estimated change as a function of substitutions in DNA sequence within ORFs. In lines 387-399 we also compare the rate of binding connection change to the rate of gene loss/gains in the four species. Although these comparisons cannot be used to assess selective constraint, it allows us to place the rate of change in the network in context of the other measures of evolutionary change in the species analyzed.

Regarding the randomizations to test whether the connections between the regulators are prevalent (previously lns. 407 – 410), we think that the test does not contribute significantly to our general conclusions and therefore have removed it from the current version of the manuscript.

Reviewer #2:This manuscript comprises an impressive breadth of experiments, yet the data is easy to follow and the figures are understandable synthesis of an enormous amount of data. The biofilm conditions selected are comprehensive and resulting gene expression data well validated and statistically supported. There are major formatting errors in Suppl Figure 4 , and minor errors in Suppl Figure 1 (top legend) and Suppl Figure 2 ( species names below blank ?).

As mentioned above in the response to Reviewer 1, this is probably an issue related to the files provided to the reviewers since the figures in the files that we are able to download from the *eLife* system are properly formatted. We will make sure that the figures are properly formatted in the proof review.

Reviewer #3:In this work, Mancera and colleagues investigate the transcriptional circuitries that control biofilm development in four medically-important *Candida* species using functional genomics. They characterize a subset of regulators that were already studied in *Candida albicans* and *Candida parapsilosis* in major publications by the groups involved in this study (Nobile et al. 2012, Holland et al. 2014) in two additional related species from the *Candida* lineage, namely *Candida tropicalis* and *Candida dubliniensis*. Although authors attempted to include some additional species from the *Candida* and *Saccharomyces* lineages in their analyses, they are faced with the complexity of the environmental cues/specificities that trigger biofilm formation in these more distantly-related organisms, reflecting the challenge of studying and explaining the evolution of some traits over extended evolutionary time scales. Still, authors provide interesting clues as to how the transcriptional circuitry that controls biofilm development operates in the four phylogenetically-close species being compared. This study could have been more impactful and comprehensive if some technical issues had been addressed/resolved (e.g. failure of tagging/expressing a subset of transcription factors and getting ChIP-seq data in the species being analyzed/compared) or if some non-useful data (e.g. other CTG species, *S. cerevisiae*/*C. glabrata*) had been removed and conclusions tempered (see some examples from specific comments).Specific comments:– Authors should rather focus on biofilm development in the 4 *Candida* species, because conditions have been optimized for the set of 4 species. For those distantly-related species, the conditions and biofilm development might not be similar at all. For instance, *S. cerevisiae* does make biofilms but in quite different ways and under different conditions/stimuli (e.g.#1, mat formation, e.g.#2, glucose inhibits adhesion and biofilm formation in *S. cerevisiae*, see PMID: 32054862).

This is an important point that we now bring up in the manuscript. However, we do think that, despite its limitations, the phenotypic comparison with other species is valuable. It shows that the phenotype is unique to these species either because of the biofilm forming abilities of the four species or due to other factors, such as the way they respond to the growth conditions. Therefore, and as suggested by the editor, we have included discussion of the issue in the corresponding Results section (ln. 254).

– Lines 189-207, description of results but no data are shown. This section does not bring much information and the conclusions are overstated (e.g. "the greater the phylogenetic distance from C. albicans, the thinner the biofilm formed") because in distantly-related species the conditions conducive to biofilm formation may not be similar to those optimized for C. albicans, C. dubliniensis, C. tropicalis and C. parapsilosis. Same for lines 210-217.

We have included a supplementary figure (Figure 2—figure supplement 2) that shows the data described in the section regarding biofilm formation by the different species studied. As mentioned above, we have also included discussion about the limitations of the comparison in the same section (ln. 254).

– Lines 234-236: "Of all the species studied (…) to physical manipulation (results not shown)". This is again an overstatement. Authors optimized biofilm conditions based on experiments performed on only 4 closely-related species.

We have included discussion about this limitation in the corresponding Results section (ln. 254).

– Figure 2: C. tropicalis CSLM data are missing in Panel B.

Due to space restrictions and the fact that biofilms formed by *C. tropicalis* are quite similar to those formed by *C. albicans* and *C. dubliniensis*, we initially decided not to show the corresponding CSLM micrograph for this species in Figure 2. We now show the *C. tropicalis* micrographs in the new Figure 2—figure supplement 2 and in Figure 3—figure supplement 1, and include this information in the figure legend of Figure 2 (ln. 1071).

– Figure 3, Supp Figure 3 and lines 270-271: "All seven master regulators identified in C. albicans were also required for biofilm formation in C. dubliniensis" – However this does not appear to stand true for C. dubliniensis BRG1. Any explanations?

Although the results of the dry-weight assay of Figure 3 do not show a clear difference between the biofilms formed by the BRG1 mutant and the WT strain, the CSLM micrographs in Figure 3—figure supplement 1 show a clear biofilm formation defect of this mutant. We have included a note in the legend of Figure 3 (ln. 1091) where we refer to the CSLM results shown in Figure 3—figure supplement 1.

– Figure 3 and lines 271-284: Data from C. albicans and C. dublinienesis should also be shown in Figure 3 to serve as a reference/benchmark. The conclusion pertaining to this figure still needs to be tempered (i.e. lines 286-287, "with the degree of diversity roughly paralleling their evolutionary distance from C. albicans"). No clear evidence supports such a conclusion, because observations were made based on mutant phenotypes from only 4 closely-related *Candida* species.

The data for *C. albicans* and *C. parapsilosis* (the data for *C. dublinienisis* is already shown) are not shown because these results were previously generated with slight variations to the assays employed (different media, for example), such that a side by side comparison would not be easy to interpret. At the end of the legend of Figure 3 (ln. 1096), we cite the corresponding references for the *C. albicans* and *C. parapsilosis* data.

The concluding sentence has been slightly modified as suggested by Reviewer 1 (ln. 314). We do think that our results support the notion that, among the four species we analyzed, the function of the TFs in the species that are further away phylogenetically from *C. albicans* are less conserved in biofilm formation.

– Lines 304-309: This is a major weakness in this manuscript. First, did author test the functionality of the tagged transcription factors (TFs)? What about their expression level in biofilm-stimulating conditions? The fact that authors failed to ChIP some of them might be due to epitope tagging which could have altered their function. Which TFs failed to be tested? It is not clearly stated in the manuscript. Again, failure to clearly show which regulators have been successfully ChIPed and the lack of data from those regulators weakens the manuscript. Maybe authors should give them a second try (by tagging differently C-term vs N-term/testing the functionality?)

We did test the functionality of the tagged TFs by performing biofilm formation assays. All tagged strains formed biofilms that could not be distinguished from the biofilms formed by the WT strains; we have added a note in the Methods section to that respect (ln. 900). Since we did not observe a defect in the tagged strains, we did not perform further analyses on them, such as measuring the expression level of the tagged regulators.

We have included the TFs for which the ChIP-seq experiments did not work in Sup Table 2 (now Supplementary File 1b). We have expanded the legend of the table so that it is easier to understand what the different values in the table mean, including the experiments that failed or that were not attempted (ln. 1170). We did perform experimental variations in the ChIP-seq experiments in an attempt to map the binding sites of the regulators for which the standard ChIP-seq experiments did not work. For example, for the *C. parapsilosis* regulators, we performed the experiment under planktonic conditions, but with no success. As the reviewer suggested, we also considered using other epitopes for tagging. However, we decided not to proceed with this strategy since it complicates the comparison to other DNA binding results as the antibodies and ChIP-seq conditions that would be used are different, bringing their own biases. In addition, since we needed to tag 27 regulators, using a variety of tags to optimize the ChIP-seq for each regulator was not a viable option. It is important to point out that tagging the strains and ensuring that they work for ChIP-seq is quite time and resource consuming; the only way to know whether the tagged regulator can be precipitated for ChIP-seq in a new species is by actually tagging and performing the ChIP-seq experiment.

Most importantly, our major conclusions would most likely not change with the addition of the DNA binding data from the regulators that were not precipitated successfully. As this reviewer noted below, most of our DNA binding data comes from Efg1 and Ndt80, both of which we were able to successfully ChIP in all four species. Therefore, the general patterns that we observed would most likely hold true with the addition of other binding data from other regulators ChIPed in a selection of species.

– Supp Table 2 is a key table and should be included in the main text. Legend is missing, it is not clear what "N/A" vs "0" stand for? It appears that Ndt80 and Efg1 targets account for the majority of binding events in the species being studied. Unlike the other regulators, these are major regulators for many important processes in *Candida* species including, for instance, filamentous growth. Consequently, the high connectivity of the biofilm network would not be surprising if one takes the Ndt80/Efg1 network as an example. This has been also shown for the Sfl1/Sfl2 transcriptional circuitry that controls filamentous growth in C. albicans (PMID: 23966855).

We agree that Sup Table 2 (now Supplementary File 1b) is important as it shows technical results of the work. Since we would like to keep the main figures and tables of the paper to show biological findings, we think this table should be kept as supplementary. As suggested, we have expanded the legend of the table so that the meaning of its content is better explained (ln. 1170).

Regarding the comment about the high connectivity of the network given that it involves Ndt80 and Efg1, we agree and include discussion of this in the Discussion section together with the Sfl1/Sfl2 reference (ln. 729). We have also incorporated this reference where we talk about Ndt80 and Efg1 working together (ln. 459).

– Figure 4B/C: It is not advised to show only percentages, as we could have the impression that we are dealing with big numbers whereas in reality we are not (e.g. Rob1, only 20 targets). Same for lines 340-343: the low number of Rob1 targets should prevent authors from drawing strong conclusions.

We have added the gross number of target genes to Figure 4 B and C and also to the whole paragraph where the overlaps of the regulators are described (ln. 356).

– Lines 345-355 and 369-385: These sections should rather go to the Discussion section or should be rewritten as data originating from experiments (i.e. results per se, not discussion). In many occurrences, authors discuss their data in the Results section. This strongly alters the quality of the reading flow. Same for lines 404-408, appearing as data but could also be moved to the Discussion section.

The three sections describe different analyses of the binding connection data and place the results in context with previous findings. Therefore, we prefer not to move them to the Discussion.

As suggested by this Reviewer and also Reviewer 1, we have moved some sections to the Discussion to improve the flow of the Results.

– Lines 415-441 and Supp Figure 4: This section is rather technical in nature and should have been presented (or summarized) earlier in the manuscript (may be following the ChIP-seq section, lines 290-343). Still, Ndt80 is not a good example for performing robustness analyses with regard to the specific conditions under which biofilm formation was induced, because this major regulator appears to exert pleiotropic functions.

As suggested, we have moved this section after the ChIP-seq section (ln 403). The major robustness analysis that we performed was the inclusion of the gene expression and binding motif data to the analysis of the binding connections. We did this for the two regulators for which we were able to identify a binding motif, Efg1 and Ndt80. Both of these regulators are involved in other cellular functions. However, our data show that their binding connections are not under very strict evolutionary constraints; they change very rapidly even between closely related species. Therefore, the robustness of their binding connections is not granted a priori.

[Editors' note: further revisions were suggested prior to acceptance, as described below.]

The manuscript has been improved but there are some remaining issues identified by one of the two reviewers that need to be addressed, as outlined below:1. Regarding the new text added on page 21-22, lines 506-570 (there is a jump from 506 to 560), I understand that the goal is to determine whether trans-changes, specifically the DNA binding specificity, could explain part of the observed binding target divergence. However, I can't quite follow the text, as it is not clear to me which species' motif was used to predict in which species' genome, and how were the overlaps actually calculated. I did look for additional information in the Methods section and couldn't find any.

To improve clarity of these analyses we have rewritten the section (ln. 468) and added description of the methodology employed in the Methods section “*de novo* sequence motif discovery and enrichment for the regulators” (ln. 928).

2. Regarding the new text from lines 619-632: am I correct in that the new results were meant to determine the relationship between binding and gene induction *in each species*, rather than attributing gene induction *differences between species* to differences in TF binding in those species? The reason for the question is because I came in expecting answers to the latter question and got confused for a moment.

That is correct. We have added the phrase “in each species” to the section to make it clearer (ln. 552). We have also tested the association of regulator binding differences with gene expression changes between species and have included the results at the end of the same Results section (ln. 567).

3. In lines 646-649, the authors laid out the challenges involved in between-species comparison, which I fully agree with. But I don't think the Ca-Cd hybrid experiment can address that. It does address a different question, i.e. specifically revealing cis-changes behind gene expression divergence, which I feel is different from the first one.

We have added the phrase “to a large extent” to lower the tone of our claim (ln. 593). We do think that the ChIP-seq experiment in the hybrid addresses most of the problems associated to species having different physiological responses to the same external environment, at least for Ndt80. In the hybrid, Ndt80 from *C. albicans* is subject to the same upstream stimuli and signaling pathways as Ndt80 from *C. dubliniensis*, and therefore the differences in regulator binding have to be mostly due to differences in the regulators binding.

4. In Figure 6A, I wonder if the authors can comment on the concave shape of the point cloud on the left.

This is an interesting observation. It would seem that Ndt80 from *C. dubliniensis* has a higher occupancy in most of the intergenic regions of the genome, but the trend reverts for the genes that are overall most strongly bound by Ndt80. Unfortunately, without further experimentation we think it would be difficult to establish a mechanistic explanation for the observation.

5. In Figure 6B, there are more than one data points for the two binding and one expression change datasets in the middle and right time points. Are those biological replicates?

For the conservation of master regulators there are three estimates because comparisons were performed between *C. albicans* and each of the other three species. On the other hand, there are six estimates of binding target and gene expression conservation since comparisons were performed in pairs between all four species. We have explained this in the corresponding figure legend (ln. 1085).

6. In the Discussion section, the authors stated that this work examined how a complex transcriptional network underlying a specific phenotype (biofilm formation) evolved over a span of ~70 million years. I think it would be useful to point out, the authors did this to some extent later, that to fully reconstruct the evolutionary history of this network, it is critical to identify all the regulators in the other three species, and that the data in this work constitutes a partial picture for the three non-albicans species.

We have included a sentence in the Discussion to clarify that further work will be needed to identify all biofilm regulators in *C. dubliniensis* and *C. tropicalis* to have a more complete picture of the network in these two species (ln. 648). In *C. parapsilosis*, a screen of gene deletion mutants to find biofilm regulators has already been performed and we do mention the regulators found in these analyses throughout the manuscript.